DOI: 10.1038/s41467-018-05729-w　　**OPEN**

# Differentiation-state plasticity is a targetable resistance mechanism in basal-like breast cancer

Tyler Risom[1], Ellen M. Langer[1], Margaret P. Chapman[2], Juha Rantala[3], Andrew J. Fields[1], Christopher Boniface[1], Mariano J. Alvarez[4], Nicholas D. Kendsersky[1], Carl R. Pelz[1], Katherine Johnson-Camacho[1], Lacey E. Dobrolecki[5], Koei Chin[6], Anil J. Aswani[7], Nicholas J. Wang[6], Andrea Califano[4,8], Michael T. Lewis[9], Claire J. Tomlin[2], Paul T. Spellman[1,10], Andrew Adey[1], Joe W. Gray [6,10] & Rosalie C. Sears[1,6,10]

Intratumoral heterogeneity in cancers arises from genomic instability and epigenomic plasticity and is associated with resistance to cytotoxic and targeted therapies. We show here that cell-state heterogeneity, defined by differentiation-state marker expression, is high in triple-negative and basal-like breast cancer subtypes, and that drug tolerant persister (DTP) cell populations with altered marker expression emerge during treatment with a wide range of pathway-targeted therapeutic compounds. We show that MEK and PI3K/mTOR inhibitor-driven DTP states arise through distinct cell-state transitions rather than by Darwinian selection of preexisting subpopulations, and that these transitions involve dynamic remodeling of open chromatin architecture. Increased activity of many chromatin modifier enzymes, including BRD4, is observed in DTP cells. Co-treatment with the PI3K/mTOR inhibitor BEZ235 and the BET inhibitor JQ1 prevents changes to the open chromatin architecture, inhibits the acquisition of a DTP state, and results in robust cell death in vitro and xenograft regression in vivo.

[1] Department of Molecular and Medical Genetics, Oregon Health & Science University, 3181 SW Sam Jackson Park Road L103, Portland, OR 97239, USA. [2] Department of Electrical Engineering and Computer Sciences, University of California at Berkeley, 253 Cory Hall #1770, Berkeley, CA 24720, USA. [3] Misvik Biology, Karjakatu 35 B, FI-20520 Turku, Finland. [4] DarwinHealth Inc., 3960 Broadway, Suite 540, New York, NY 10032, USA. [5] Lester and Sue Smith Breast Center, Baylor College of Medicine, One Baylor Plaza – BCM600, Houston, TX 77030, USA. [6] Center for Spatial Systems Biomedicine, Oregon Health & Science University, 2730 SW Moody Avenue, Portland, OR 97201, USA. [7] Department of Industrial Engineering and Operations Research, University of California at Berkeley, 4141 Etcheverry Hall, Berkeley, CA 94720, USA. [8] Department of Systems Biology, Irving Cancer Research Center, Columbia University, 1130 St. Nicholas Avenue, 8th Floor, New York, NY 10032, USA. [9] Department of Biochemistry and Molecular Biology, Baylor College of Medicine, One Baylor Plaza – BCM125, Houston, TX 77030, USA. [10] Knight Cancer Institute, Oregon Health & Science University, 3181 SW Sam Jackson Park Road, Portland, OR 97239, USA. These authors contributed equally: Margaret P. Chapman, Juha Rantala. Correspondence and requests for materials should be addressed to J.W.G. (email: grayjo@ohsu.edu) or to R.C.S. (email: searsr@ohsu.edu)

The mammary gland contains a diverse repertoire of epithelial cell states that rely on chromatin dynamics for specification[1,2]. Throughout development, these states include distinct fetal and adult stem cell states, lineage-restricted luminal and myoepithelial progenitors, mature luminal and myoepithelial states, and mesenchymal-transitioned cells[3–7]. While DNA methylation plays a predominant role in early lineage distinction in the maturing embryo[8], cell differentiation from stem cell states in the adult is primarily carried out through dynamic changes in histone modifications at promoters and distal regulatory elements[2,9,10], altering the open chromatin architecture and providing enhanced expression of new lineage and differentiation genes[11,12]. These chromatin dynamics are critical for the specialized cell state heterogeneity that maintains normal mammary gland function.

Tumors that arise from the complex epithelial compartment of the mammary gland are also phenotypically diverse. Many breast tumors display intratumoral phenotypic heterogeneity[13–15] and are populated with tumor cells in functionally distinct cell states. Different cell states can possess distinct drug sensitivities[15–19], making cell-state heterogeneity a challenge for therapeutic management of breast tumors. An additional challenge to therapeutic treatment is the inherent plasticity of tumor cell states[20–22]. Cytotoxic and targeted therapies have been shown to drive cells into drug tolerant persister (DTP) cell states that can survive drug pressure in a low-proliferative state[19,23,24], leading to incomplete response and/or recurrence. Recent findings demonstrate that dynamic chromatin remodeling processes, similar to those employed in normal cell fate determination, can underlie these transitions to drug-tolerant states[24–26]. While it is well established that Darwinian selection of genetically diverse cellular subpopulations[27,28] can contribute to therapeutic resistance, mounting evidence implicates chromatin remodeling as another critical driver of resistance[24–26,29]. Understanding which breast tumor subtypes have high cell state heterogeneity and propensity for cell-state plasticity, whether specific therapeutics trigger DTP transitions, and what targetable epigenomic processes underlie these transitions will be critical steps to improving management of heterogeneous breast tumors.

Here, we use an operational metric of differentiation-state heterogeneity to identify breast tumor subtypes with high intratumoral heterogeneity, and then use models of these subtypes to investigate how cell-state heterogeneity and plasticity contribute to the generation of DTP cell states. We identify multiple classes of targeted therapeutics that steer initially heterogeneous cell populations to more homogeneous but persisting states and use gene expression profiling to identify upregulated signaling and epigenetic pathway activity in the DTP cells. We show through genome and epigenome analysis, as well as mathematical modeling, that the development of drug persisting populations occurs primarily through epigenomic transition and not Darwinian selection of preexisting resistant subpopulations. Through analysis of transcriptional profiles of drug persisting populations, we find BRD4 activity is upregulated in the DTP cells following treatment with MEK or PI3K/mTOR targeted therapies. We demonstrate that combination treatment with JQ1, an inhibitor of bromodomain and extraterminal (BET) family proteins including BRD4, can prevent the global change in open chromatin architecture that accompanies DTP state formation during PI3K/mTOR inhibitor response. Moreover, combination of PI3K/mTOR and BET inhibitors drives complete cell kill of basal-like breast cancer cell lines (BCCLs) in vitro, and tumor regression of orthotopic xenografts in vivo. Our study demonstrates that triple-negative (TN) and basal-like breast cancers show high cell-state heterogeneity and plasticity, and advances our understanding of how drug combinations targeting chromatin

dynamics can improve management of these aggressive subtypes of breast cancer.

## Results

**Differentiation-state heterogeneity in breast cancer.** We examined the relationship between differentiation-state heterogeneity and breast tumor subtype in primary patient samples, patient-derived xenografts (PDX), and BCCLs. We measured single cell expression of Cytokeratin 19 (K19), Cytokeratin 14 (K14), and Vimentin (VIM) to define distinct differentiation states, as these intermediate filament markers are preferentially expressed in the luminal, myoepithelial (basal), and mesenchymal cell states of the normal breast, respectively[4,6,30,31]. We first profiled treatment-naïve tumors of varying hormone receptor status, including "luminal" (ER+/PR+/HER2−), HER2+ (ER+/−/PR−/HER2+) and triple-negative (TN, ER−/PR−/HER2−) tumors, using immunofluorescent imaging and image cytometry and examining multiple tumor regions if possible (Fig. 1a, Supplementary Data 1). We identified single nuclei using a DNA counterstain (DAPI) and assessed cytoplasmic expression of K19, K14, and VIM in an expanded region around each nucleus (Supplementary Fig. 1a, b). All epithelial marker positive cells (K19+ or K14+) were considered to be tumor-derived, excluding histologically determined normal structures and in situ lesions. We determined the frequency of differentiation states within each tumor region and calculated the diversity of these states using the Shannon diversity index[32]. We found that luminal and HER2+/ER+ tumors were almost exclusively composed of a K19+/K14−/VIM− differentiation-state while a subpopulation of K19+/K14+/VIM− cells was observed in some HER2+/ER− tumors (Fig. 1b; Supplementary Fig. 1c; Supplementary Data 1). In contrast, most TN tumors contained numerous K19, K14, and VIM-defined differentiation states, including robust proportions of epithelial tumor cells expressing more than one of these differentiation-state markers. The mean Shannon diversity index was, therefore, significantly higher in TNBC tumors than in non-TN tumors (Fig. 1c). We next examined 31 molecularly-profiled PDX breast tumors[33] of varying hormone receptor status and molecular subtype (Fig. 1d, e, Supplementary Fig. 1d, e, Supplementary Data 1). We used human specific antibodies to identify the additional tumor populations of K19−/K14−/VIM+ and K19−/K14−/VIM− cells, and observed robust heterogeneity within these PDX models (Fig. 1e). Consistent with our observations in primary patient samples, the Shannon diversity index was significantly higher in TN tumors compared to non-TN tumors (Fig. 1f). Further, tumors of the basal-like molecular subtype had significantly higher Shannon diversity indices than the HER2 molecular subtype tumors (Fig. 1g). Analysis of IF-stained BCCLs (Fig. 1h, Supplementary Fig. 2a, b, c) supported our observations in primary and PDX tumors: TN cell lines had significantly higher diversity indices than non-TN lines, and both TN molecular subtypes (basal-like and claudin-low) showed significantly higher diversity indices compared to luminal B and HER2 subtype BCCLs (Fig. 1i, j). Cell line heterogeneity was maintained in 3-dimensional culture and orthotopic xenografts of these cell lines (Supplementary Fig. 2d).

We examined the expression of luminal, basal, and mesenchymal markers in RNAseq data from BCCLs[34]. We focused on the expression of 25-gene genesets for each lineage, including genes specific to the luminal or myoepithelial differentiation states of the normal breast, as well as 25 classic epithelial-to-mesenchymal transition markers[2,4–7,30,31,35,36]. We performed unsupervised clustering on 44 BCCLs using these 75 genes (Supplementary Fig. 2e). Claudin-low lines clustered with dominant mesenchymal gene expression. Luminal B lines, as well as the majority of HER2

lines, clustered with dominant expression of luminal genes. Basal-like lines expressed myoepithelial-specific (basal) genes, but also showed high levels of luminal and mesenchymal gene expression (Fig. 1k). We calculated the cumulative *Z*-score mean of, as well as variance between, the luminal, myoepithelial, and epithelial–mesenchymal transition (EMT) genesets to serve as metrics of molecular differentiation-state heterogeneity. Basal-like

lines showed significantly higher cumulative *Z*-score means and significantly lower variances between the genesets compared to all other groups (Fig. 1l m, Supplementary Fig. 2f). Of note, there was a biphasic distribution of the variance metric within HER2 cell lines, which correlated with a newer molecular classification of these lines as L-HER2 and HER2E subtypes[37,38] (Fig. 1n). The metric of variance between genesets was inversely correlated with

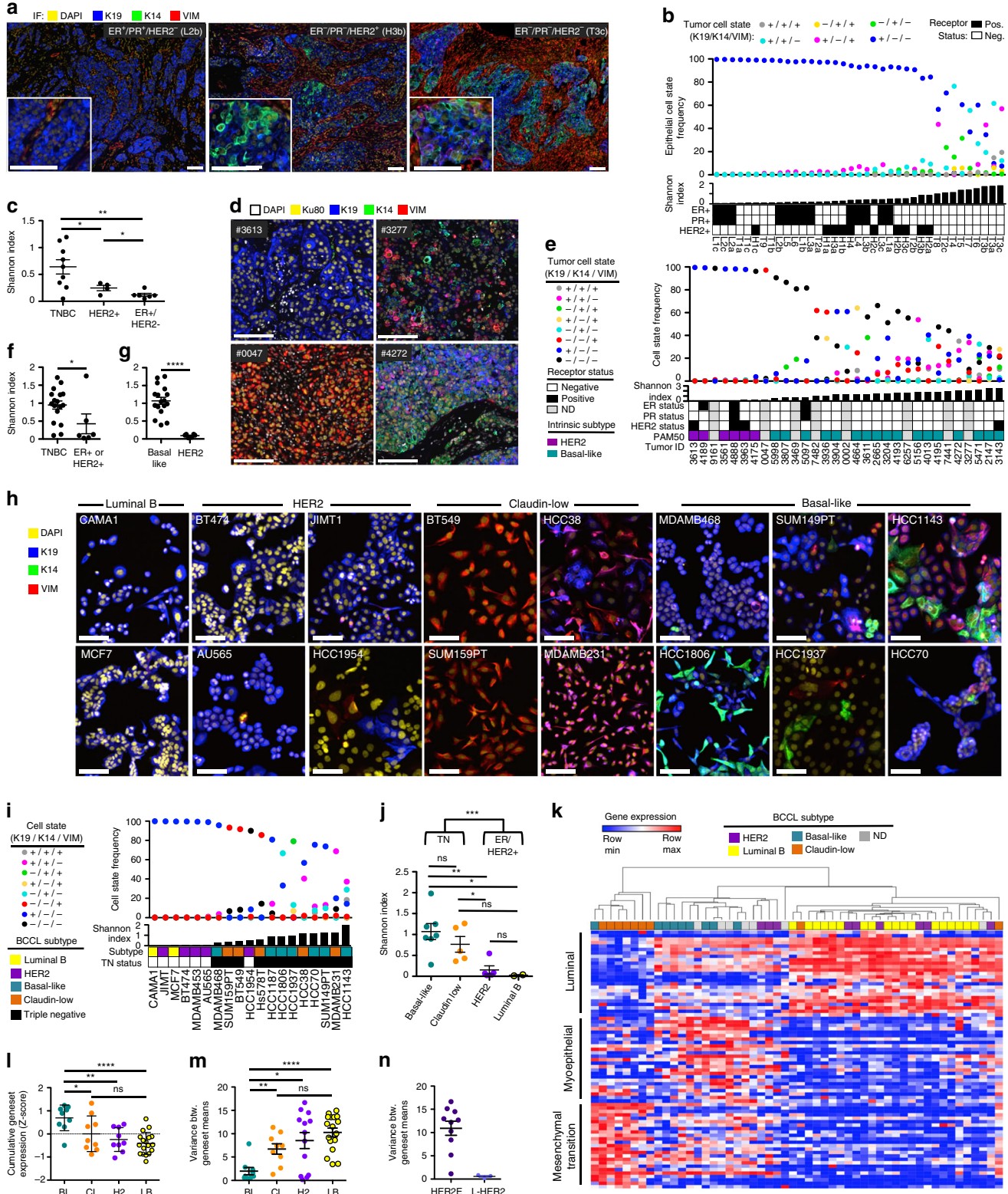

the Shannon diversity indices calculated from the cell line imaging data (Supplementary Fig. 2g).

Together these data show that TN tumors, and more specifically the basal-like molecular subtype, harbor the most differentiation-state heterogeneity among breast tumors. Importantly, this heterogeneity was maintained in basal-like BCCLs, suggesting these lines can serve as models of how phenotypic heterogeneity affects therapeutic response.

**Targeted therapies generate distinct DTP cell states**. We investigated the impact of targeted therapies in the heterogeneous basal-like cell lines, HCC1143 and SUM149PT, following a 72 h treatment with seven doses of each of 119 pathway-targeted therapeutic compounds. We quantified changes in cell number and single cell expression of K19, K14, and VIM by analysis of immunofluorescent images (Fig. 2a, Supplementary Fig. 3, Supplementary Data 2). Most drugs had incomplete cytotoxicity at maximum doses and exhibited altered differentiation-state marker expression in the DTP cells. We used K-means clustering to identify compounds that produced DTP cells with similar patterns of K19/K14/VIM expression. Figure 2a shows six general response groups in HCC1143 representing DTP cells with: (A) increased K14 expression; (B) increased K19 and VIM expression; (C) increased expression of all markers at high dose; (D) increased expression of all markers across doses; (E) minimal change in marker expression; or (F) variable, non-dose-dependent response. Importantly, compounds with the same molecular target or related pathway targets clustered together. For example, inhibitors targeting MEK and BRAF grouped together within the K14-enriched cluster (Group A), inhibitors targeting mTOR and PI3K grouped within the K19/VIM-enriched cluster (Group B), inhibitors targeting the ErbB receptors, Src family kinases, or Aurora kinases increased expression of all markers (Groups C and D), and inhibitors of Flt3 and other related kinases had minimal influence on cell number or differentiation-marker expression (Group E). Results in SUM149PT cells showed a similar array of phenotypic responses, with drug targets clustering into groups with similar marker expression as that seen in HCC1143 (Supplementary Fig. 3, Supplementary Data 2).

We further analyzed compounds associated with the K14-enriched and K19/VIM-enriched groups since these left cells in distinct states. We evaluated responses of HCC1143 cells after 72 h treatment with escalating doses of two MEK inhibitors,

Trametinib and AZD6244 (K14-enriched cluster, Group A), and two dual-specificity PI3K/mTOR inhibitors, BEZ235 and PI103 (K19/VIM-enriched cluster, Group B). As observed in the initial screen, these agents left DTP populations with distinct differentiation marker expression (Fig. 2b). The MEK inhibitors produced DTP cells with large cytoplasmic volume, high K19 and K14 expression, and reduced VIM expression. Conversely, the PI3K/mTOR inhibitors produced DTP cells with high K19 and VIM expression and reduced K14 expression. The divergent effects of MEK and PI3K/mTOR inhibition on DTP differentiation-state were also observed in SUM149PT cells and could be seen in both lines stained for additional basal (Cytokeratin 5 (K5) and Cytokeratin 17 (K17)) and luminal (Cytokeratin 8 (K8)) markers (Supplementary Fig. 4a, b). MEK and PI3K/mTOR inhibitors both produced large DTP populations indicated by high projected maximal inhibition "Einf" values and a plateau in the dose response curve (Fig. 2c, Supplementary Fig. 5a). MEK inhibition drove dose-dependent increases in cellular mean-fluorescent intensities (MFI) of K14 and K19, while PI3K/mTOR inhibitors caused mean cell MFI reductions in K14 and increases in K19 and VIM. To determine the conservation of these phenotypic responses to therapy, we assayed eight basal-like cell lines with diverse genetic backgrounds and different baseline differentiation-state heterogeneities (see Fig. 1h) for change in mean-cell MFI of K8, K19, K14, K5, and VIM following treatment with low and high dose of Trametinib and BEZ235 (Fig. 2d). Unsupervised clustering of phenotypic responses to these agents showed that the majority of these basal-like lines shared a similar phenotypic response: Trametinib enriched a K19/K5/K14-high basoluminal state with lower K8 and VIM levels, and BEZ235 enriched a state marked by low basal cytokeratin expression and increased levels of K19 expression in most lines, and K8 in some lines. Assessment of the phenotypic response to these drugs in luminal B and claudin-low cell lines showed that Trametinib and BEZ235 affected cell proliferation, but the cells remained in their respective K19$^+$/K14$^-$/VIM$^-$ and K19$^-$/K14$^-$/VIM$^+$ differentiation states following treatment (Supplementary Fig. 5b).

We further assessed differentiation-state enrichments following MEK or PI3K/mTOR inhibition by analyzing treated cell populations using RNA-sequencing and geneset enrichment analysis (GSEA) for 32 curated genesets relating to normal breast cell states[2,31,39], breast cancer subtypes[40–43], and breast cancer

---

**Fig. 1** Differentiation-state heterogeneity is enriched in the triple negative and basal-like subtypes. **a** Representative IF images of treatment-naïve primary breast cancers: "luminal" (ER$^+$/PR$^+$/HER2$^-$), HER2+ (ER$^{+/-}$/PR$^-$/HER2$^+$), and triple negative (ER$^-$/PR$^-$/HER2$^-$), scale bars = 100 μm. **b** The frequency of six epithelial cell states is shown for each tumor region in a vertical scatterplot with accompanying Shannon diversity index and ER, PR, and HER2 status. Luminal (L), HER2+ (H), and TN (T) tumor regions are arranged left to right by increasing Shannon index, regions of the same tumor denoted by "a, b, c" e.g., L2a, L2b, L2c. **c** Graph comparing Shannon index between tumors of different hormone receptor subtype (multiple regions of individual tumors are averaged if available), asterisks denote significance *$P < 0.05$, **$P < 0.01$. SEM shown. **d** IF images of PDX tumors with low (left panels) and high (right panels) Shannon indices stained for DAPI (white), Ku80 (yellow), K19 (blue), K14 (green) and VIM (red), scale bars = 100 μm. **e** The frequency of eight tumor cell states based on K19, K14, and VIM expression is shown for 31 PDX tumors in a vertical scatterplot with accompanying Shannon index. Patient ER, PR, and HER2-receptor status and intrinsic molecular subtype is shown. Tumors arranged left to right by increasing Shannon index. **f, g** Graphs of Shannon index, comparing tumors with differing receptor-positivity status and molecular subtype. *$P < 0.05$, ****$P < 0.0001$. SEM shown. **h** IF images of BCCLs with differing molecular subtypes, scale bars = 100 μm. **i** The frequency of eight cell states based on K19, K14, and VIM expression is shown for each BCCL in a vertical scatterplot with accompanying Shannon index, molecular subtype and Triple negative (TN) status is indicated (denoted by color, marked by black squares). Cell lines are arranged left to right by increasing Shannon index. **j** Graph of Shannon index, comparing TN and non-TN BCCLs, as well as different molecular subtypes. *$P < 0.05$, **$P < 0.01$, ***$P < 0.001$, ns = not significant, SEM shown. **k** Heatmap of BCCL gene expression of 25 luminal, 25 myoepithelial, and 25 EMT-correlated genes. BCCLs are arranged by unsupervised clustering. Cell line molecular subtype is denoted by color. **l** Graph of the cumulative Z-score of the luminal, myoepithelial, and EMT genesets in BCCLs of different molecular subtype, basal-like (BL), claudin-low (CL), HER2+ (H2), and Luminal B (LB), *$P < 0.05$, **$P < 0.01$, ****$P < 0.0001$, ns = not significant, SEM shown. **m** Graph of the variance between the mean geneset expression of the luminal, myoepithelial, and EMT genesets in BCCLs of different molecular subtype, asterisks denote significant difference in geneset variance, SEM shown. **n** Graph of the variance between HER2+ cell lines that are either of the L-HER2 or HER2E molecular subtypes

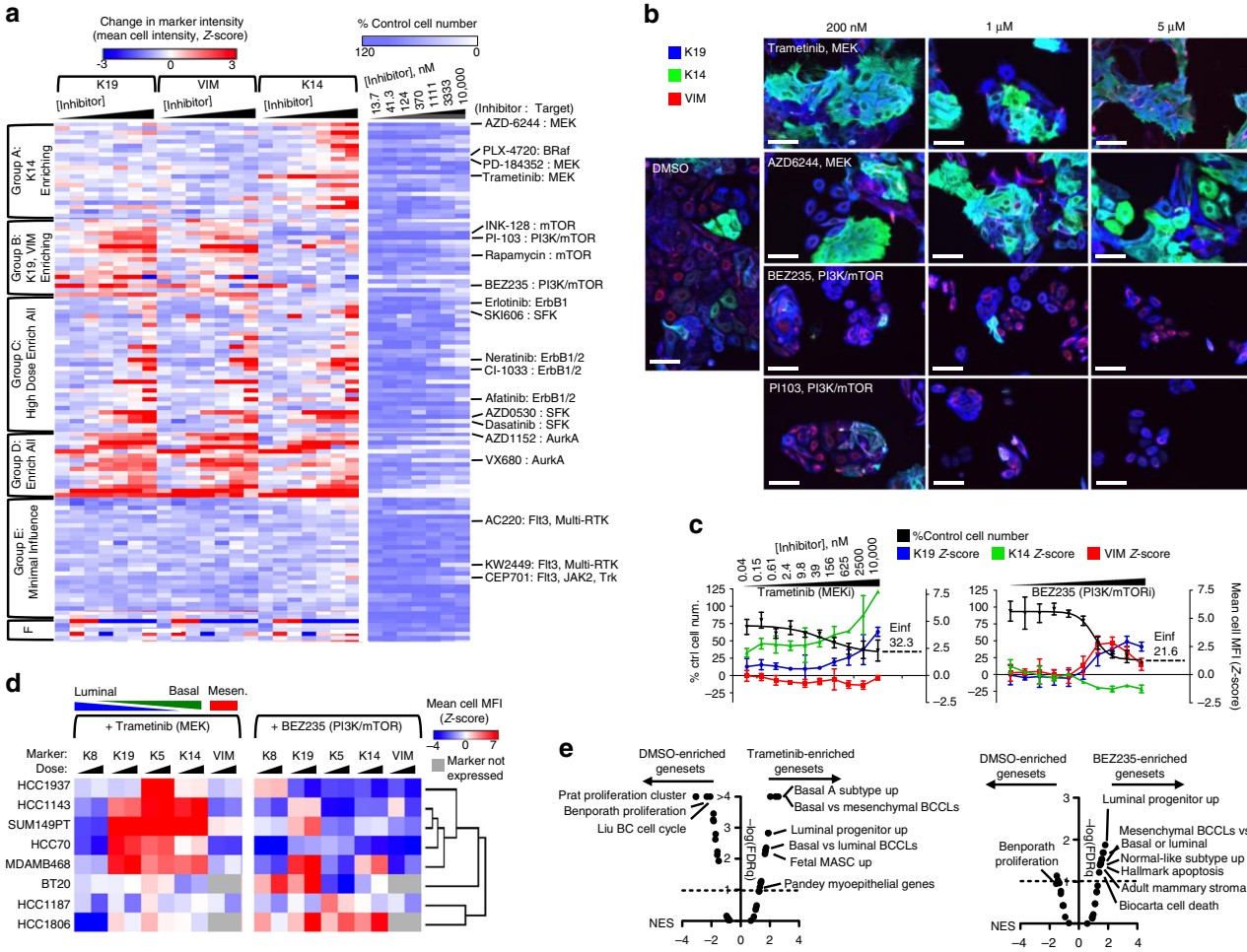

**Fig. 2** Targeted therapies enrich distinct drug-persisting differentiation states. **a** Heatmaps show the change in K19, VIM, and K14 expression compared to DMSO control wells as a Z-score (left), and change in proliferation (right, percent of control) of HCC1143 cells following 72 h exposure to seven doses of 119 targeted therapeutics. K-means clustering on differentiation marker expression shows six distinct clusters. Select drugs from each cluster and their primary therapeutics target(s) are labeled to the right. A full list of drugs and responses is in Supplementary Dataset 2. **b** IF images of HCC1143 cells following 72 h treatment with increasing doses of MEK inhibitors Trametinib and AZD6244 and the PI3K/mTOR inhibitors BEZ235 and PI103, or a DMSO control, showing K19 (blue), K14 (green), and VIM (red), scale bars = 100 μm. **c** Graphs of therapy-induced changes in cell number (black, left axis) and mean-cell MFI (right axis, as Z-score) of K19 (blue), VIM (red), and K14 (green), in HCC1143 cells following 72 h incubation with increasing doses of Trametinib or BEZ235. The projected maximum level of inhibition, or Einf, is shown for each drug. **d** Heatmaps of the change in mean-cell MFI (Z-Score) for K8, K19, K5, K14, and VIM is shown for eight basal-like BCCLs following 72 h incubation with 250 or 2500 nM Trametinib (left) or BEZ235 (right). Unsupervised clustering, using data from both agents, was used to group cell lines based on phenotypic response. Markers for luminal, basal, and mesenchymal phenotype are indicated. **e** GSEA results as a volcano plot of Normalized Enrichment Score (NES, x-axis) vs. FDRq (−log, y-axis) examining 32 genesets related to mammary cell states and breast cancer subtypes, enrichment compared between DMSO and 1 μM Trametinib (left), or DMSO and 1 μM BEZ235 (right) in HCC1143 cells treated for 6 days. Select top-enriched genesets are labeled

proliferation[40,44] (Supplementary Data 3 "Breast Phenotype Genesets"). In Trametinib-treated cells, we observed significant enrichment of genesets specific to the Basal A subtype of breast cancer[41], basal BCCLs[42], and myoepithelial cells of the normal breast[31] (Fig. 2e, Supplementary Fig. 5c). These results are consistent with the increased basal cytokeratin expression we observed following Trametinib treatment. We also observed enrichment of genesets specific to human luminal progenitor cells[39] and murine fetal mammary stem cells (fMASC)[39], two related progenitor/stem states that show dual expression of basal and luminal markers[3,4,39], consistent with the observed enrichment of the K19hi/K14hi differentiation-state following Trametinib treatment. In BEZ235 DTPs, we observed enrichment of genesets relating to mesenchymal BCCLs[42], the normal-like subtype of breast cancer[43] and adult mammary stroma[39], in addition to enrichment of the luminal progenitor geneset[39]. This

was consistent with the reduction of basal cytokeratin expression and mixed expression of luminal and mesenchymal markers in these cells (Supplementary Fig. 5d). Finally, both treatments induced significant de-enrichment of proliferation genesets, and BEZ235 treatment resulted in enrichment of apoptosis and cell death genesets (Supplementary Data 3).

**Cell state dynamics during therapy.** We next explored the dynamics of therapy-induced differentiation-state changes and the roles of state-selective proliferation, state-selective cell death, and cell-state transition in the phenotypic response to targeted therapy. We defined four prominent differentiation states as (1) cells that expressed high levels of K14, "K14hi"; (2) cells that express high levels of VIM and low levels of K14, "VIMhi"; (3) cells that express high levels of K19 and low levels of K14 and VIM, "K19hi"; and (4) cells that express low levels of all markers, "K19low/

VIM^low/K14^low" (with "high" defined as a cell with MFI exceeding the average MFI in DMSO wells and "low" as an MFI below the DMSO average, Supplementary Fig. 6a). We measured the frequency of these states every 12 h following treatment with Trametinib or BEZ235 in HCC1143 cells. Trametinib induced a four-fold increase in the frequency of the K14^hi state (27–81%) while significantly reducing the frequency of the other three states over time compared to a DMSO control (Fig. 3a). Conversely, BEZ235 induced a significant increase in the K19^hi state and a robust reduction of the K14^hi state over time. We tested whether differential growth of pre-existing K14^hi and K14^low states under drug pressure could explain the observed state enrichments. We pulsed cells with 5-ethynyl-2′-deoxyuridine (EdU) prior to

fixation at each 12-h timepoint and found that both drugs had a strong cytostatic influence, reducing EdU-incorporation to near zero with Trametinib, and to one-third of control levels with BEZ235 (Fig. 3b). Cells remained in this low proliferative state as long as drug pressure was maintained (up to 21 days, Supplementary Fig. 6b). Importantly, no significant differences in S-phase frequency were observed between cells with high or low expression of K14, K19, or VIM, except for VIM^hi vs. VIM^low rates 24 h post-Trametinib treatment, which we attributed to noise due to the low frequency of this population under Trametinib treatment (Fig. 3b, c). Mass cytometry experiments supported these observations, showing only minimal differences in IdU incorporation between cells with high and low expression of

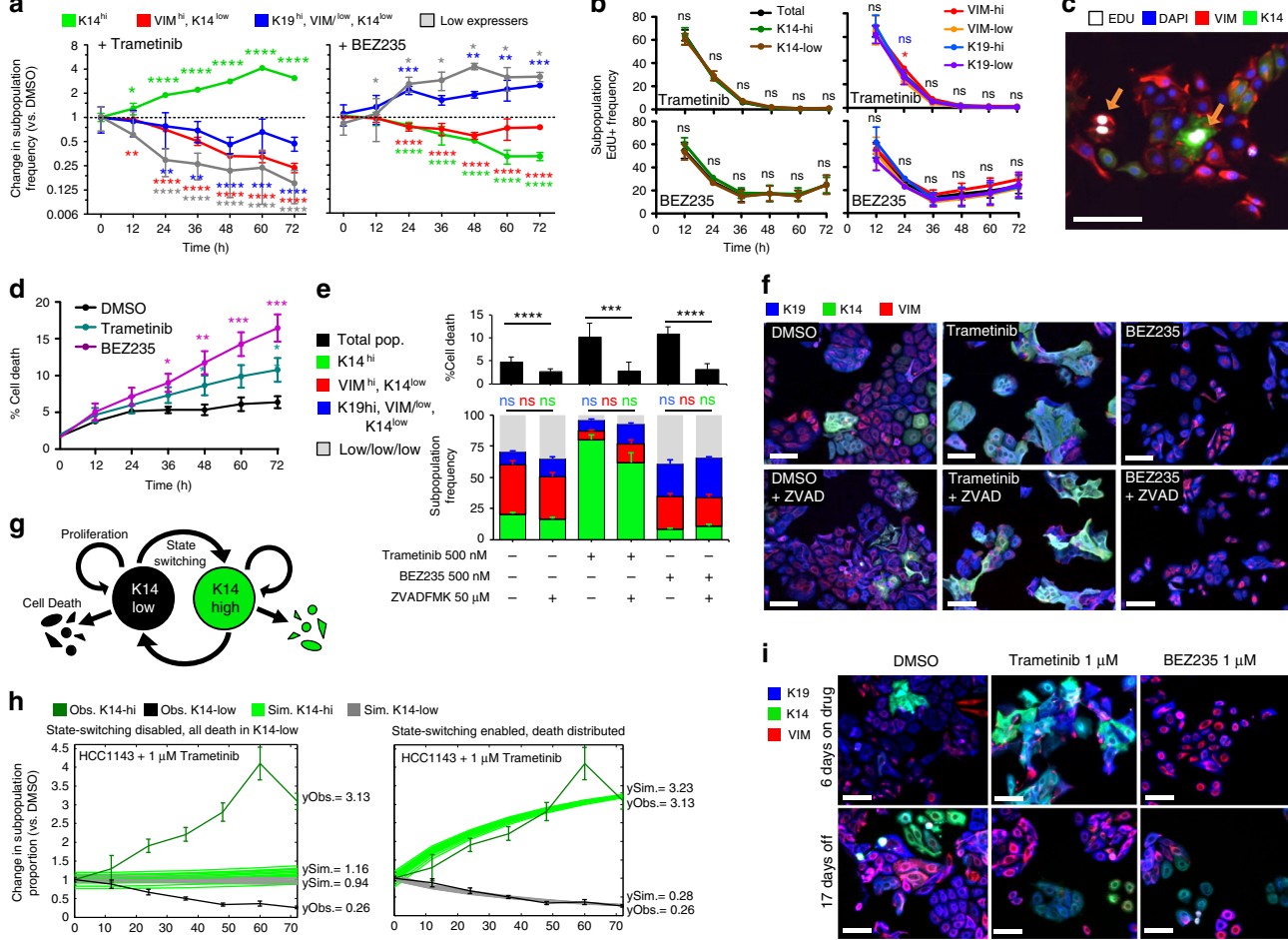

**Fig. 3** Cell state transitions underlie DTP-state enrichment. **a** Graphs showing change in frequency (vs. DMSO) of four differentiation states defined as K14^hi (green), VIM^hi/K14^low (red), K19^hi/VIM^low/K14^low (blue), and K19^low/VIM^low/K14^low (gray) following exposure to 1 μM Trametinib or BEZ235. Asterisks depict significant gains or losses in state frequency vs. 0 h, *P < 0.05, **P < 0.01, ***P < 0.001, ****P < 0.001, n = 15 with SD. **b** Graphs showing percent of EdU+ cells expressing high or low levels of K14, K19, or VIM following treatment with 1 μM Trametinib or BEZ235, ns = not significant, SEM shown. **c** IF image showing both K14^hi and K14^low cells positive for EdU-incorporation following 36 h of Trametinib treatment, scale bar = 100 μm. **d** Graph showing percent of YO-PRO-1+ dying cells following treatment with 1 μM Trametinib or BEZ235, n = 8 with SEM. **e** Graphs showing the percent of dying cells (top) and the frequency of cells in four differentiation states (bottom, defined and colored as in (**a**)) following 72 h exposure to Trametinib, BEZ235, or DMSO +/− the pan-caspase inhibitor Z-VAD-FMK. No significant (ns) differences observed in differentiation-state frequency +/− Z-VAD-FMK, n = 12 with SD. **f** IF images of remaining HCC1143 cells in (**e**) following 72 h exposure to Trametinib, BEZ235 or DMSO +/− Z-VAD-FMK. Scale bars = 100 μm. **g** Schematic of cell-state behavior where cells can transition between K14^hi and K14^low cell states and undergo death or proliferate in either state. **h** Simulated fold change of K14^hi (bright green) and K14^low (gray) cell-state proportions over 72 h following treatment with 1 μM Trametinib (vs. DMSO) are overlaid with experimentally observed average fold change (vs. DMSO) of K14^hi (dark green) and K14^low (black) cell-state proportions. ySim denotes simulated endpoint values; yObs denotes observed endpoint values. Two simulations with different cell-switching and cell death parameters shown: *K14^hi Darwinian selection* (left, Trametinib kills only K14^low cells, and cell-state transition is inhibited) and *Transition-mediated* (right, Trametinib kills K14^low and K14^hi cells in equal proportions, cell-state transition is allowed). **i** IF images of HCC1143 cells following 6 days of 1 μM Trametinib or 1 μM BEZ235, then following 17 days of culture without drug. scale bars = 100 μm

luminal and basal markers following Trametinib or BEZ235 treatments (Supplementary Fig. 6c-e). Since all differentiation states appeared to have similar cytostatic responses to each therapy, we investigated whether cell death contributed to differentiation-state enrichments following Trametinib or BEZ235 treatments. We examined total cell death throughout the 72 h treatment by measuring the incorporation of YO-PRO-1 into treated cells and found that BEZ235 induced significant gains in cell death beginning at 36 h while Trametinib only showed significant gains in cell death compared to control at 48 and 72 h (Fig. 3d). Notably, significant gains in cell death occurred after significant changes in differentiation-state frequencies were observed (see Fig. 3a). We were limited in our ability to examine state-specific cell death due to non-specific antibody staining in dying cells and the loss of adherent-cell morphology. As an alternative approach, we tested the necessity of cell death for the observed phenotypic outcomes by combining Trametinib or BEZ235 with Z-VAD-FMK, an inhibitor of caspase-mediated cell death. Z-VAD-FMK treatment significantly reduced cell death in all conditions (Fig. 3e). However, the drug-induced differentiation-state composition at 72 h for these cells was nearly identical to that for cells treated with Trametinib or BEZ235 alone (Fig. 3e, f).

We next examined whether genomic selection occurs in the drug-induced differentiation-state enriched DTPs. We performed whole exome sequencing on HCC1143 cells following exposure to Trametinib, BEZ235, or DMSO. We first conducted a pair-wise examination to identify single nucleotide variants (SNVs) as well as insertions or deletions (indels) that differed between any of the treatments (Supplementary Table 1). Despite robust phenotypic enrichments following Trametinib and BEZ235 (Supplementary Fig. 7a), and good overall sequencing coverage (Supplementary Fig. 7b, c), we did not observe results consistent with Darwinian selection. The variants and indels identified between treatment conditions (annotated list in Supplementary Table 1, vcf files available in Supplementary Data 4) had low variant allele frequencies (VAF) (highest 26%, avg. 13%), substantially lower than the fraction of cells showing the enriched phenotypes (>70%) following Trametinib or BEZ235 treatment. All variants (26/26) and the majority of indels (17/19) involved intronic, intergenic, or non-coding genomic locations and were predicted to have no functional impact on gene products. In addition, we had low confidence in the majority of these calls due to technical issues (Supplementary Table 1). We looked further into two indel calls, one that identified a potential frameshift deletion in CD93, present in both DMSO- and Trametinib-treated cells, and one that identified a potential deletion in the 3′UTR of TRMT1L, present in both BEZ235- and Trametinib-treated cells. Both of these indel calls were located in highly repetitive genomic regions, suggesting potential alignment artifacts. Further, gene expression analysis revealed that CD93 was not expressed in this cell line and TRMT1L showed no significant changes in gene expression following treatment (Supplementary Fig. 7d). We next identified tumor-specific SNVs and indels in our WES data through comparison of each condition to a patient-matched normal sample (HCC1143-BL) sequenced in a previous study[34]. We identified ten variants and three deletions that were common to all conditions with very similar VAFs between the three treatments (annotated list in Supplementary Table 2, vcf files available in Supplementary Data 4). Six SNVs were identified in one or two conditions as compared to normal, but similar to the pairwise comparison above, these were all in intronic or intergenic regions not predicted to affect protein function and all had low VAFs (highest 13%, avg. 10%). Finally, we also performed low-pass whole genome sequencing (WGS) using the same DNA from control, Trametinib, and BEZ235-treated

HCC1143 cells. The copy number profile of HCC1143 matched that of a previously study[45], and showed remarkable concordance between the drug-treated groups (Supplementary Fig. 7e, segmentation files available in Supplementary Data 4). Together these results suggest that genomic selection plays a minimal role in the observed differentiation-state enrichments by Trametinib and BEZ235.

We also explored the potential for state-selective death or cell-state transition to contribute to drug-induced differentiation-state enrichment using a cell-state dynamical model of proliferation, death, and transition between K14$^{hi}$ and K14$^{low}$ states (Fig. 3g, Supplementary Methods). Cell-state proliferation rates were set to be equal since that was measured directly with our EdU+ analysis (see Fig. 3b). We modeled the impact of state-specific selection of K14$^{hi}$ cells, by allowing cell death following Trametinib only from the K14$^{low}$ state and not allowing cell-state transition. Conversely, we modeled transition-mediated enrichment by allowing cell death following Trametinib from both K14$^{hi}$ and K14$^{low}$ states and permitting transitions between cell-states. The state-specific selection model showed very little change in K14$^{hi}$ and K14$^{low}$ subpopulations over time, which was inconsistent with our experimental observations (Fig. 3h, left). However, the cell-state transition model was consistent with the experimental observations following Trametinib treatment (Fig. 3h, right). In simulations of BEZ235 response, the transition-mediated model also was more consistent with the observed changes in K14$^{hi}$ and K14$^{low}$ state frequencies (Supplementary Fig. 8a). Finally, if enrichment of specific cell states occurred through state transitions, we would expect that these therapy-induced states would be reversible and cells would return to heterogeneity following withdrawal of the drugs. Indeed, we observed control-levels of heterogeneity within 17 days of drug withdrawal, and also found these cells retained their pre-treatment sensitivity to BEZ235 and Trametinib (Fig. 3i, Supplementary Fig. 8b-d). In sum, our experimental data and mathematical models support cell state transitions as the driver of DTP-state aggregation following Trametinib and BEZ235 treatment rather than Darwinian selection.

**Drug combinations reduce therapeutic escape**. We analyzed DTP cells following BEZ235 and Trametinib treatment to determine whether cells in states enriched by these treatments rely on specific pathways for survival. We measured gene expression profiles using RNA-seq in DTP cells after 6 days of treatment with Trametinib or BEZ235 and used the Virtual Inference of Protein-activity by Enriched Regulon (VIPER) algorithm[46] to identify differential activity of regulatory proteins between BEZ235- and Trametinib-treated HCC1143 cells compared to a DMSO control. We then mapped the VIPER-inferred protein activity signatures onto multiple pathway-ontology databases using the DAVID functional annotation tool[47] (Supplementary Fig. 9a, Supplementary Data 5). These analyses suggested that cells remaining after treatment with a PI3K/mTOR inhibitor had increased pathway activities including MAPK, BCL-2, and NFkB, while those remaining after treatment with a MEK inhibitor had increased pathway activities including PI3K, integrin, FGF, and JNK (Supplementary Fig. 9b). Both DTP populations shared enrichment of JAK-STAT, Notch, TGF, and Ras pathway regulators. We identified upregulated and targetable proteins in each of these pathways, and evaluated the anti-proliferative synergy of drug combinations designed to counter the upregulated pathways. BEZ235 was tested in combination with ABT737 (BCL2 mimetic) and SCH772984 (ERKi), Trametinib was tested in combination with SP600125 (JNKi) and EHOP-16 (RACi), and both drugs were tested in combination with TG101384 (JAK2i) and DAPT (γ-secretase

inhibitor). Finally, we tested the combination of BEZ235 and Trametinib. Drugs were scored as synergistic if they had combination indices[48] below 1 at 75% (CI75) and 90% (CI90) inhibitory doses in HCC1143 and SUM149PT cells (Fig. 4a, Supplementary Fig. 9c-e). Synergy was observed in most of the combinations predicted by pathway activity. Direct combination of BEZ235 and Trametinib resulted in the best combination indices, significantly increased cell death (Fig. 4b), and significantly reduced S-phase as indicated by reduced EdU-incorporation rates (Fig. 4c). Despite these encouraging metrics of drug synergy, 100% cell death was never obtained, except with EHOP-16, which showed high single agent toxicity (Fig. 4d). We assessed the differentiation states of DTP cells treated

with the combination of Trametinib and BEZ235 in three basal-like cell lines, HCC1143, SUM149PT, and HCC70, by immuno-fluorescence staining for K19, K14, and VIM. These analyses showed that all three cell lines were enriched for a K19hi state after treatment with Trametinib + BEZ235 (Fig. 4e, Supplementary Fig. 10a). We measured the frequency of the four differentiation-marker defined cell states (as in Fig. 3a) over time in response to Trametinib + BEZ235 in HCC1143 cells and found a significant time-dependent increase in the K19hi state and decrease in the VIMhi state. The K14hi state, which included K19hi/K14hi cells, did not change over time (Fig. 4f). GSEA results showed that gene expression changes following Trametinib + BEZ235 were consistent

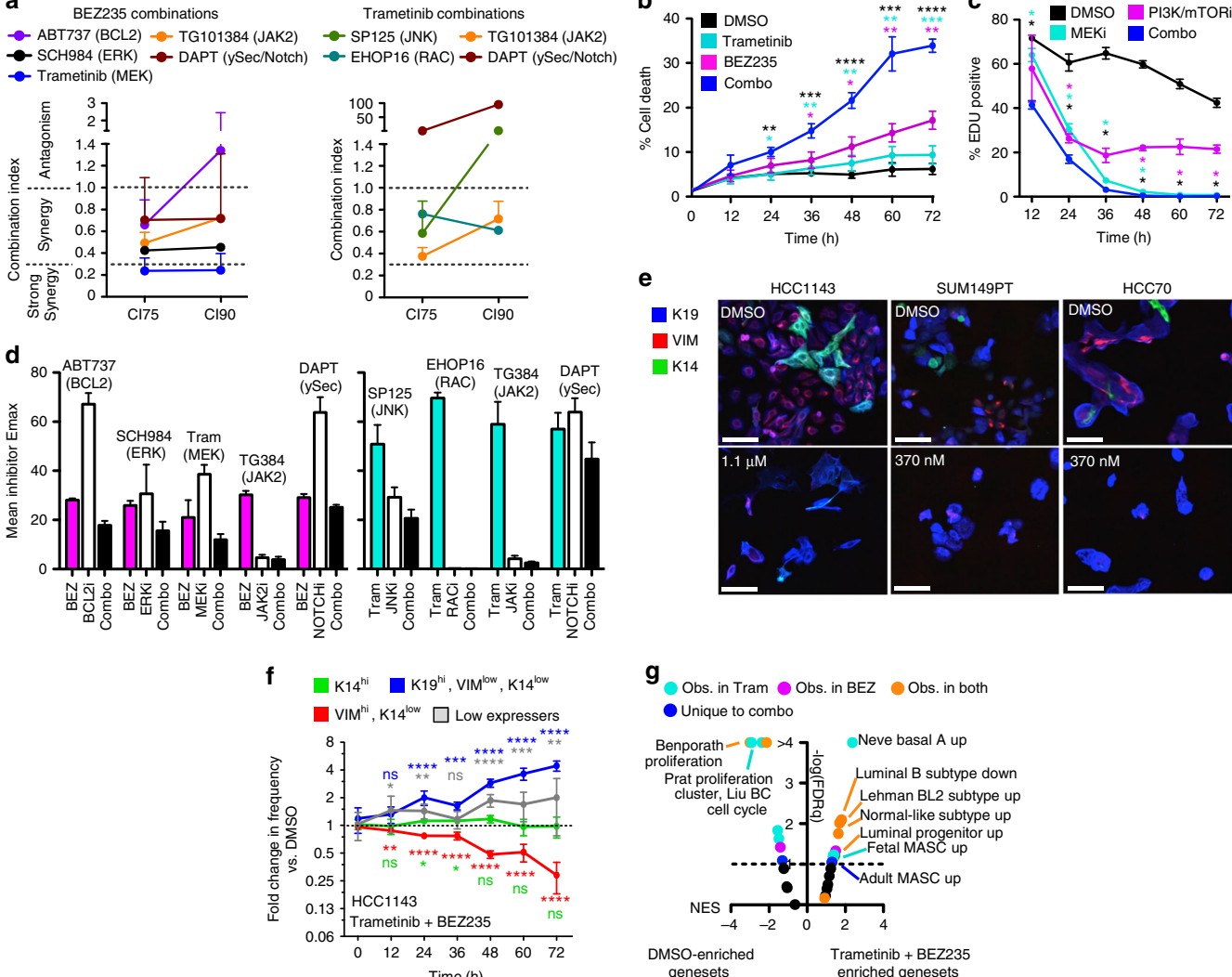

**Fig. 4** Drug combinations targeting DTP-enriched pathways still leave persisting cells of distinct identity. **a** Graphs of combination indices (CI) from drug combinations including BEZ235 (left) or Trametinib (right) and agents targeting upregulated pathway regulators in DTP states identified by VIPER analysis. CIs were calculated at 75% (CI75), and 90% (CI90) dose inhibitory values from replicate colorimetric proliferation assays, $n = 3$ with SEM. **b** Graph showing the percent of YO-PRO-1+ dying cells in HCC1143 cells following the addition of 1 μM BEZ235, 1 μM Trametinib, the combination of the two drugs, or a DMSO control. Asterisks denote significant gains in percent cell death in combination-treated cells, *$P < 0.05$, **$P < 0.01$, ***$P < 0.001$, ****$P < 0.001$, $n = 4$ with SEM. **c** Graphs show the percent of EDU+ HCC1143 cells following treatment with 1 μM Trametinib, 1 μM BEZ235, a combination of agents, or DMSO control. Asterisks denote significant reduction in %EdU+ in combination-treated cells, $n = 4$ with SD. **d** Graphs compare the maximal inhibition (Emax, as % control proliferation) for single agent BEZ235, Trametinib, the agents from (**a**), or combinations thereof. $n = 3$ with SEM. **e** IF images of three basal-like cell lines before and after 72 h exposure to the indicated doses of BEZ235 + Trametinib, DAPI not shown, scale bars = 100 μm. **f** Graphs show the fold change in frequency (vs. DMSO) of four differentiation states in HCC1143, as in Fig. 3a, b, following addition of 1 μM BEZ235 + 1 μM Trametinib. Asterisks depict significant gains or losses in state frequency vs. 0 h, $n = 15$ with SD. **g** GSEA results as a volcano plot of Normalized Enrichment Score (NES, x-axis) vs. FDRq (−log, y-axis) examining 32 genesets related to mammary cell states and breast cancer subtypes, enrichment compared between DMSO and 1 μM Trametinib/BEZ235 combination in HCC1143 cells treated for 6 days. Select top-enriched genesets are labeled

with decreased proliferation and that genesets enriched by combination treatment were a mixture of those observed with either single agent, with few genesets uniquely enriched by this treatment (Fig. 4g, Supplementary Data 3 "Phenotype 6day COMBOvsDMSO"). The GSEA results along with the minimal change in K14 expression suggest that the two drugs counteract their divergent effects on the myoepithelial differentiation program and leave cells in a state of reduced proliferation. In this DTP state, the combination-treated cells showed decreased sensitivity to numerous FDA-approved cytotoxic therapies (Supplementary Fig. 10b). Similar to the single agent treatments, acquisition of this DTP state did not involve state-selective cytostasis, cells could remain in this DTP state with minimal proliferation for at least 3 weeks, and this state was reversible following withdrawal of the drugs and subsequent cultures regained combination sensitivity (Supplementary Fig. 10c-f).

**Inhibiting differentiation-state transitions with BET inhibition.** Since we determined that the DTP states arise primarily

through state transitions but failed to find a targeted combination that achieved 100% cell kill, we explored the possibility of inhibiting chromatin modifications as a therapeutic strategy to make state-transition-inducing drugs more effective. We used GSEA to assess the expression of genes associated with chromatin modification enzyme activity in Trametinib- or BEZ235-treated DTP cells and found that genesets in DTP cells after both treatments suggested increased BRD4[49,50], KDM5B[51], and EZH2[14] activity (Fig. 5a, Supplementary Data 3 "Chromatin 6day BEZvsDMSO"). We tested the possibility that BET protein-mediated chromatin regulation supports drug-induced transitions by combining the BET inhibitor JQ1 with Trametinib and BEZ235. The combination of JQ1 and BEZ235 resulted in complete cell kill in all four basal-like cell lines, preventing the formation of a DTP population, although treatment with JQ1 or the kinase inhibitors as single agents did not (Fig. 5b). Dose–response curves for JQ1 + BEZ235 treatment showed significant increases in maximal inhibition (Emax) and negative projected maximal inhibition (Einf) values for all basal-like lines tested (Fig. 5b, c), but not for

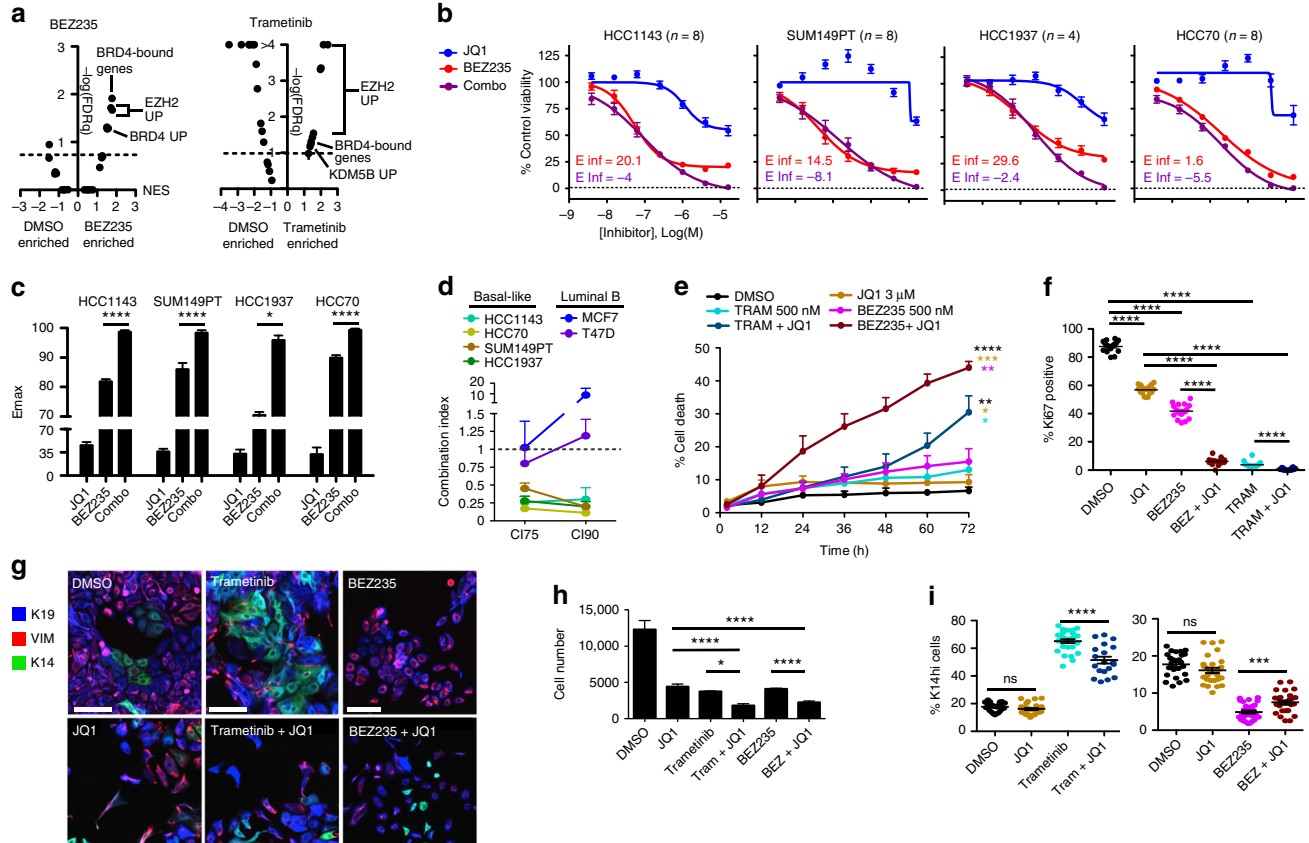

**Fig. 5** BET inhibitor combinations improve cell kill and suppress DTP transition. **a** GSEA results as a volcano plot of Normalized Enrichment Score (NES, x-axis) vs. FDRq (−log, y-axis) examining 25 chromatin modifier enzyme activity-related genesets, enrichment compared between DMSO and 1 µM BEZ235 (left), or DMSO and 1 µM Trametinib (right) in HCC1143 cells treated for 6 days. Select top-enriched genesets are labeled. **b** Dose–response curves show the efficacy of BEZ235 alone (red), JQ1 alone (blue), or a combination of the two agents (purple, equimolar ratio) in four basal-like cell lines using a colorimetric proliferation assay. E-infinity (Einf) values of single agent BEZ235 (red) and BEZ235 + JQ1 (purple) are displayed. n = 4–8 with SEM. **c** Maximal inhibition (Emax) by the agents shown in (**b**) is displayed for each basal-like cell line, asterisks depict significant gains in Emax with JQ1 + BEZ235 compared to BEZ235 alone, *P < 0.05, **P < 0.01, ***P < 0.001, ****P < 0.0001. **d** Graph showing combination indices for the BEZ235 + JQ1 drug combination at 75% (CI75), and 90% (CI90) inhibitory doses for four basal-like and two luminal B BCCLs, n = 5 with SEM. **e** Graph showing the percent of dying cells (YO-PRO-1+) in HCC1143 following the addition of 1 µM BEZ235, 1 µM Trametinib, 2 µM JQ1, the combinations of these agents, or a DMSO control. Asterisks denote significant gains in percent cell death in combination-treated cells vs. single agent BEZ235 or Trametinib, n = 3 with SEM. **f** Graph showing the percent of Ki67 positive HCC1143 cells following 72 h of DMSO, 2 µM JQ1, 1 µM Trametinib, 1 µM BEZ235, or combinations with JQ1. Asterisks denote significant difference in Ki67+ frequency, n = 16 with SD. **g** IF images of HCC1143 cells following 72 h exposure to DMSO, 400 nM BEZ235, 400 nM Trametinib, 8 µM JQ1, or the combination of these agents, scale bars = 100 µm. **h, i** Graphs show total cell number and the frequency of K14hi cells following the treatments outlined in (**e**). Asterisks denote significant change in the frequency of K14hi cells, or cell number, n = 27 with SD

Luminal B cell lines (Supplementary Fig. 11a, b). Analyses of these dose–response curves showed strong synergy between JQ1 and BEZ235 in all basal-like lines, resulting in CI75 and CI90 values <0.5, whereas these drugs had an antagonistic relationship in Luminal B lines with CI75 and CI90 values near or above 1 (Fig. 5d). Trametinib + JQ1 also produced synergistic CI values and increases in Emax in basal-like cell lines, however, the combination was not able to completely kill HCC1143 cells at maximum dosing, lacked synergy at CI90, and had positive Einf values, suggesting an inability to produce complete cell kill (Supplementary Fig. 11c-e). The triple combination of Trametinib, BEZ235, and JQ1 also produced complete cell kill and prevented DTP formation in HCC1143 (Supplementary Fig. 11f). This synergistic relationship between JQ1 and the MEK and PI3K/mTOR inhibitors was also consistent with observed significant gains in cell death (Fig. 5e) and significant reduction of proliferation (Fig. 5f).

We also analyzed the influence of JQ1 on differentiation-state marker expression in HCC1143 cells treated at sub-lethal doses with Trametinib, BEZ235, JQ1, or combinations thereof. JQ1 combined with either Trametinib or BEZ235 significantly suppressed differentiation-state transitions. JQ1 + Trametinib maintained a significantly lower frequency of the K14$^{hi}$ cell state, and JQ1 + BEZ235 maintained a significantly higher frequency of the K14$^{hi}$ state as compared to the single treatment conditions (Fig. 5g–i).

**JQ1 prevents chromatin accessibility changes associated with DTP generation**. We used single cell combinatorial indexing Assay for Transposase Accessible Chromatin sequencing (sciA-TAC-seq[52]) to understand how BEZ235 and Trametinib affect the open chromatin architecture to induce DTP states, and to determine whether BET protein inhibition alters this process. Latent semantic indexing followed by t-distributed Stochastic Neighbor Embedding (t-SNE) was performed to visualize global chromatin architecture differences at single cell resolution between DMSO-, BEZ235-, and Trametinib-treated HCC1143 cells (Fig. 6a). This analysis revealed that BEZ235 and Trametinib change the landscape of accessible chromatin regions significantly, inducing distinct state-space enrichment in the t-SNE plots. Interestingly, small subpopulations of DMSO cells occupy the BEZ235 and Trametinib enriched regions, consistent with therapeutic enrichment of a particular chromatin architecture that can exist under normal growth conditions (Supplementary Fig. 12a). We next examined transcription factor (TF) DNA-binding-motif prominence within the open chromatin regions of BEZ235- and Trametinib-DTP cells using chromVAR[53] and found that the drugs had profound effects on the accessibility of specific TF motifs. BEZ235 DTPs showed enrichment of motifs for AP1, ATF, and TCF as well as motifs for EMT-promoting TFs ZEB1, TWIST1/2, SNAI1/2, and luminal lineage TFs ESR1, PGR and GATA3 (Fig. 6b, Supplementary Data 6). This is consistent with mixed luminal and mesenchymal gene expression programs in those cells. Trametinib DTPs showed enrichment of motifs for the SP/KLF family of TFs, Homeobox TFs, and the myoepithelial lineage-specific TF EGR1[2] and other EGR family members (Fig. 6b, Supplementary Data 6). NFkB and Rel motifs were enriched in DTPs from both treatments.

We used sciATAC-seq to further assess how BET inhibition with JQ1 suppresses DTP-state enrichment by BEZ235. Figure 6c shows that BEZ235 enriches a distinct open-chromatin state space as compared to DMSO and that combination with JQ1 inhibits this change, resulting in open chromatin architecture equivalent to single agent JQ1 and highly overlapping with the DMSO-treated population (Fig. 6c, Supplementary Fig. 12b). In

accordance with the inhibition of chromatin alterations, combination of JQ1 with BEZ235 brought open motif patterns back to near-DMSO patterns (Fig. 6d, Supplementary Fig. 13a, b, Supplementary Data 6). The prevention of changes to TF accessibility with BEZ235 + JQ1 combinations was consistent with reduced expression of TF-activity-related genesets as measured by GSEA: decreases in AP1, SNAIL, GATA3, NFKB, and TCF motif accessibility in BEZ235 + JQ1-treated cells relative to BEZ235-treated cells correlates with de-enrichment of genesets related to FRA1[54], SNAI1[55] and GATA3[56] overexpression in mammary cancer cells, NFkB activity-related genesets, and a WNT activity signature[57] (Fig. 6e, Supplementary Data 3 "TF and C6 3day BEZJQ1vsDMSO"). We further assessed GATA3 expression and found that BEZ235 drove increased *GATA3* mRNA expression and increased nuclear GATA3 protein levels, whereas Trametinib reduced these levels (Fig. 6f, Supplementary Fig. 14a-d). In addition to the effects of JQ1 combination on GATA3 motif accessibility, JQ1 prevented the increases in *GATA3* mRNA and protein expression when combined with BEZ235 (Fig. 6g, Supplementary Fig. 14e, f). JQ1 combination with BEZ235 also prevented the enrichment of breast phenotype genesets associated with the BEZ235 DTP state (Fig. 6h, for BEZ235-induced enrichments see Fig. 2e). Importantly, loss of proliferation and increased apoptosis genesets did not reverse with JQ1 combination, but were enriched in all drug-treated conditions (Fig. 6h). The ability of JQ1 to prevent DTP-specific geneset enrichment was also observed in combination with Trametinib (Supplementary Fig. 14g, Supplementary Data 3 "Phenotype 3day TRAMJQ1vsDMSO").

Since we found dramatic inhibition of BEZ235-induced state changes by JQ1 along with synergistic gains in cell death and reductions in proliferation, we tested the efficacy of the BEZ235 + JQ1 combination in vivo in an orthotopic xenograft model using the tumor-forming basal-like HCC70 cells that show similar heterogeneity, DTP states, and JQ1 effects (see Figs. 1h, 2d, 5b–d). The BEZ235 + JQ1 combination significantly reduced tumor volume compared to either single agent (Fig. 6i, Supplementary Fig. 15a), consistent with the synergy observed in vitro (Fig. 5b, d). Furthermore, immunofluorescent staining of control and treated tumors showed that while BEZ235 treatment alone decreased the number of K14 expressing cells, the combination treatment of BEZ235 + JQ1 suppressed this phenotypic shift (Fig. 6j, Supplementary Fig. 15b). These results are consistent with our observations in vitro (Fig. 5g–i), supporting the ability of JQ1 to prevent BEZ235-induced DTP transitions.

## Discussion

Our overall goal in this study was to develop therapeutic strategies to more effectively treat breast tumors that exhibit intratumoral phenotypic heterogeneity and plasticity. We used immunofluorescence staining for K14, K19, and VIM to operationally define differentiation-states and assessed their expression heterogeneity in primary breast cancers and PDX. We demonstrate that TN breast cancers and the basal-like molecular subtype harbor high levels of differentiation-state heterogeneity. BCCLs model the differentiation-state heterogeneity observed in their respective tumor subtypes, with basal-like cell lines being populated by cells coexisting in numerous differentiation-states with distinct luminal, basal, and mesenchymal gene expression. Examination of the phenotypic response of these basal-like cell lines to 119 pathway-targeted inhibitors revealed that most drugs are ineffective at inducing complete cytotoxicity in these lines, and leave residual cells with altered expression of differentiation-state markers. Basal-like cell lines have been shown to have a high propensity for differentiation-state transition under normal

growth conditions[20], and our work demonstrates that this inherent plasticity is prominent upon therapeutic challenge. Our observations also align with previous studies that show the generation of a low-proliferative, drug-tolerant persister "DTP" cell state following treatment with various targeted agents[24,25,58]. Indeed, we found that MEK and PI3K/mTOR inhibitors, as well as the combination of these agents, drove basal-like cell lines into DTP states that were marked by reduced proliferation, distinct differentiation marker expression, and could persist for weeks under high doses of therapy.

Contrary to a model of Darwinian selection, no differentiation-state specific cytostasis was evident in our experiments and cell death was not required for the observed phenotypic changes following Trametinib and BEZ235. Genomic analyses also failed to find evidence of significant genomic selection, indicating that cell-state transitions are the major driver of DTP state acquisition.

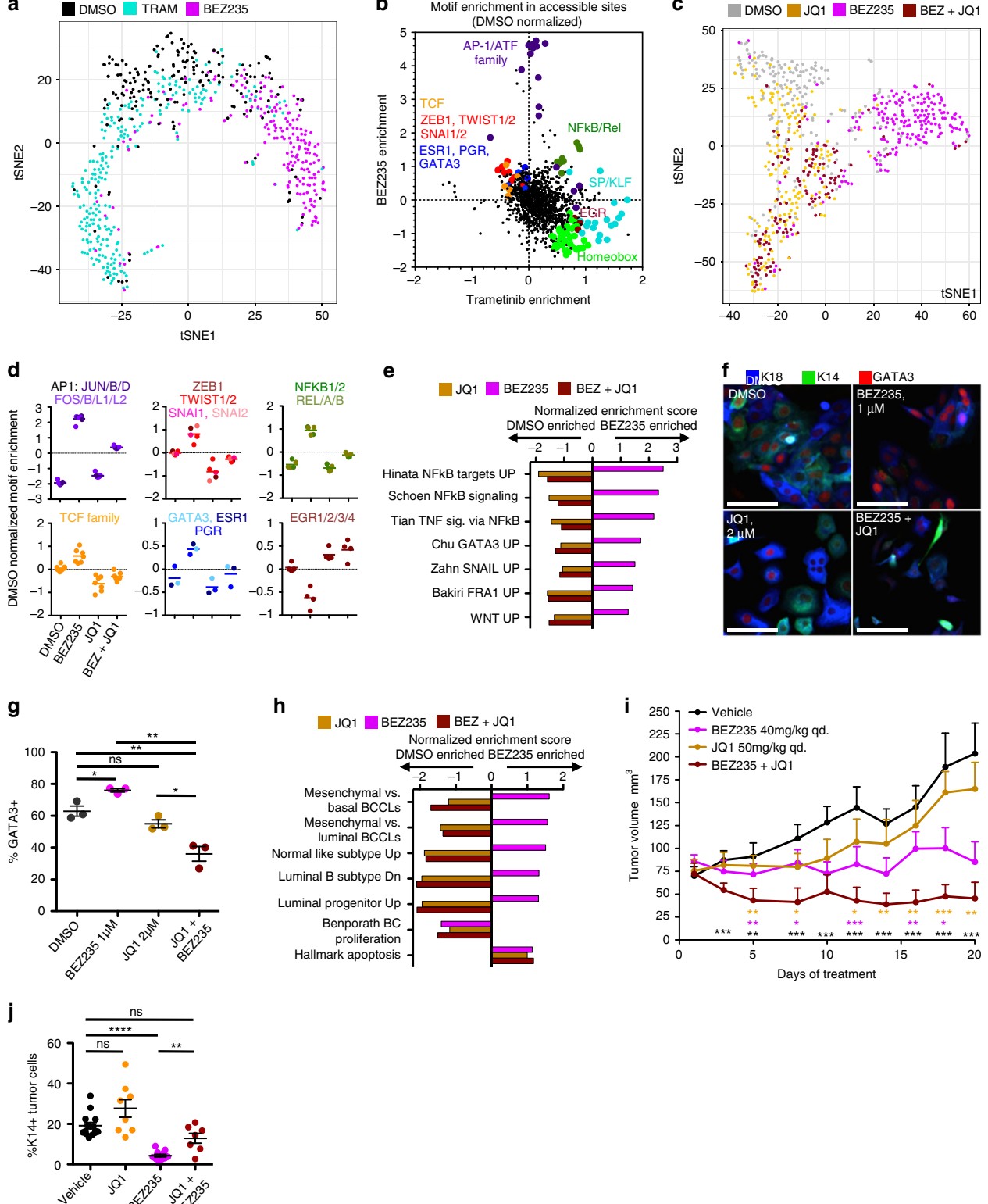

Consistent with previous observations of reversible drug tolerance[24,25,58], cells grown out of these DTP states after drug withdrawal regained the differentiation-state heterogeneity and drug sensitivity of treatment-naïve cells. Computational modeling also supported state-transitions as the driving mechanism behind generation of the observed DTP state aggregation by MEK and PI3K/mTOR inhibitors.

We show evidence that these cell-state transitions are driven by changes to the epigenome. We find that both Trametinib and BEZ235 promote increases in gene expression signatures related to epigenetic modifier enzyme activity, including BET family epigenetic reader proteins, histone demethylase KDM5B, and histone methyltransferase EZH2. Further, through the use of single cell ATAC-seq, we demonstrate that profound chromatin architecture changes underlie the phenotypic responses to MEK and PI3K/mTOR inhibition. These chromatin level changes greatly alter the transcription-factor accessibility landscape, evidenced by enrichment and depletion of known TF DNA binding motifs in open chromatin sites. Altered binding site enrichments included EGR1 in Trametinib DTPs, and AP1, ZEB1, SNAI1 and GATA3 in BEZ235 DTPs, consistent with the role of these TFs in basal or mesenchymal, and luminal differentiation, respectively, in breast cells[2,7,14,54,56]. Importantly, combination of BET inhibitors with the kinase-targeting agents suppressed chromatin remodeling and prevented the induction of these TF transcriptional programs, instead promoting robust growth inhibition and cell death.

These findings align with studies that characterize compensatory kinase upregulation in response to MEK and PI3K inhibitors[59–61]. Recently, this Trametinib-induced "kinome reprogramming" was shown to involve increased BRD4 association with kinase gene enhancers[62], allowing for the expression of new kinases that circumvented the MEK signaling blockade. Targeted BET inhibition was similarly shown to combat drug tolerance in this study. Our findings indicate that BRD4 not only supports drug tolerance by eliciting compensatory kinase gene enhancement, but also by altering the chromatin accessibility of pivotal TFs involved in both cell survival and differentiation control.

The findings of this work and related studies demonstrate that some cancers, such as TN and basal-like breast cancers, possess enhanced phenotypic and epigenomic plasticity. This plasticity enables rapid acquisition of a drug-tolerant state when these cancers are treated with pathway-targeted agents, driven through adaptive changes to their epigenome. The efficacy of small-molecule kinase inhibitors in these epigenomically-plastic tumor subtypes will therefore depend on our ability to combat these adaptive epigenomic changes. We demonstrate here that combinations using BET inhibitors paired with PI3K or MEK inhibitors represents one successful strategy to combat this process, and to elicit robust cell death and tumor regression in vivo. While BET inhibitor combinations were successful in this setting, the efficacy of targeting other epigenetic factors should be evaluated to best improve our management of tumors with high phenotypic heterogeneity and plasticity.

## Methods

**Cell lines.** All human cell lines were obtained from the American Type Culture Collection (ATCC) other than JIMT1 (DSMZ), SUM149PT (Asterand), and SUM159PT (Asterand). Cell lines were cultured according to supplier protocol with supplemental 10 μg/ml penicillin and streptomycin (Thermo) and regularly tested to ensure cultures were negative for mycoplasma. Cell line genotype was confirmed by STR profiling to ensure accurate identity. All lines were maintained at 37 °C in a 5% $CO_2$ atmosphere and cultured at a cellular confluence below 80%.

**Reagents.** Primary antibodies: Cytokeratin 19 (Dako, Clone RCK108), Cytokeratin 14 (Abcam, Clone LL002), Vimentin (Cell Signaling, Clone D21H3), Cytokeratin 5 (Abcam, Clone EP1601Y), Cytokeratin 17 (Thermo, clone E3), Claudin 4 (R&D Systems, Clone 382321), Cytokeratin 8 (K8, Abcam, Clone M20), Ku80 (Cell Signaling, Clone C48E7), Ki67 (DAKO, Clone MIB-1), GATA3 (Cell Signaling, Clone D13C9), Cytokeratin 18 (Cell Signaling, Clone DC10).

Secondary antibodies: (all LifeTech unless noted); goat-anti-mouseIgG1-Alexa647, goat-anti-mouse IgG3-Alexa488, goat-anti-mouseIgG2a-Alexa488, goat-anti-mouseIgG2b-Alexa488, donkey-anti-rabbit-Alexa568, donkey-anti-goat-Alexa647, goat-anti-rabbit-dylight755 (Thermo).

Small molecule inhibitors: All drugs, unless otherwise noted, were purchased from Selleckchem including (+)-JQ1 for in vitro experiments. BEZ235 was purchased from LC Laboratories and (+)-JQ1 for in vivo studies was provided by Jay Bradner at the Dana-Farber Institute of Harvard, Cambridge, MA. All in vitro inhibitor stocks were solubilized in DMSO and stored as 10 mM stock solutions at −80 °C.

**Image cytometry of primary tumor samples.** All samples were formalin-fixed, paraffin-embedded sections of treatment-naïve primary breast tumor samples of hormone-receptor-defined subtypes: luminal (ER$^+$/PR$^+$/HER2$^-$, $n = 6$), HER2+ (ER$^-$/PR$^-$/HER2$^+$, $n = 3$, ER$^+$/PR$^+$/HER2$^+$, $n = 1$), and triple-negative (ER$^-$/PR$^-$/HER2$^-$, $n = 9$). Tumor specimens were obtained from three sources: Tumors with multiple analyzed regions (L1-3, H1-3, T1-3) were from surgical blocks obtained from the OHSU Knight BioLibrary, five samples were core biopsy specimens (L4-6, H4, T4) obtained from the OHSU Knight BioLibrary, and five samples were from a tissue microarray of TN breast cancers surgical blocks created under IRB-approved protocols with patient consent from the University of California, San Francisco. With pathologist assistance, areas of high tumor cellularity and low immune infiltrate or stromal density were identified for the focus of immunofluorescent analysis. Cut sections of 5 μm were de-paraffinized in xylene and passed through a series of graded alcohols. Antigen retrieval was performed in a 0.1 M sodium citrate buffer pH 6 (Sigma) under heat and pressure, followed by blocking with a 5% donkey serum (Sigma), 5% goat serum (Vector Laboratories), 1% BSA (Fisher) blocking buffer. Sections were incubated overnight at 4 °C with a primary antibody solution against K19 (1:300), K14 (1:300), and VIM (1:200) diluted in 1% BSA, 2.5% donkey serum, 2.5% goat serum. Sections were washed in PBS with 0.1% Tween (Fisher) and secondary antibody staining was performed at room temperature for 1 h with AlexaFluor secondary antibodies against primary host species (1:200, LifeTech) in 1% BSA and 5% animal serum. 1 μg/ml 4′,6-Diamidino-2-phenylindole nuclear counterstain (DAPI, LifeTech) was added to secondary staining buffers. Surgical specimens were imaged on a Zeiss Axio

**Fig. 6** Changes in open chromatin architecture underlie DTP transition and are inhibited by JQ1. **a** t-SNE plot of all cells following 72 h of 1 μM BEZ235, 1 μM Trametinib, or a DMSO control in HCC1143, calculated from sciATAC-seq results. Single cells are colored based on treatment, BEZ235 (magenta), Trametinib (cyan), DMSO (gray). **b** A dot plot showing enriched DNA-binding protein motifs in the open chromatin sites following BEZ235 or Trametinib treatment, normalized to DMSO values. Select related transcription factor motifs are enlarged, colored, and labeled. **c** t-SNE plot of all cells following 72 h of 1 μM JQ1 (yellow), 1 μM BEZ235 (magenta), JQ1 + BEZ235 (maroon), or a DMSO control (gray), calculated as in (**a**). **d** Line graph showing the level of motif-enrichment for six groups of transcription factors is shown for DMSO, BEZ235, JQ1, and JQ1 + BEZ235 treatments. **e** Graph of GSEA results showing transcription-factor activity-related genesets shown to be significantly enriched ($P < 0.05$) following 72 h of BEZ235 treatment. The NES in JQ1-treated, and BEZ235 + JQ1-treated cells shown adjacently. **f** IF images showing K18, K14, and GATA3 expression in HCC1143 cells following 72 h treatment, scale bars = 100 μm. **g** Graph showing the frequency of GATA3+ cells in HCC1143 following treatment, asterisks depict significant differences in GATA3+ frequency, *$P < 0.05$, **$P < 0.01$, ****$P < 0.001$, $n = 3$ with SEM. **h** Graph of GSEA results showing the NES of breast phenotype genesets shown to be significantly enriched ($P < 0.05$) following 72 h of BEZ235 treatment, with the subsequent NES of that geneset in JQ1-treated, and BEZ235 + JQ1-treated cells shown adjacently. **i** A graph showing change in tumor volume in HCC70 xenografts treated with vehicle control (black), BEZ235 (red), JQ1 (blue), or BEZ235 + JQ1 (purple). Asterisks denote significant difference in tumor volume between combination-treated tumors and other groups, denoted by color, $n = 8$ with SEM. **j** Graph showing the frequency of K14+ human (KU80+) tumor cells in each HCC70 xenograft following treatment with BEZ235, JQ1, the combination of agents, or vehicle control, $n = 8$ with SEM

Microscope capturing 3-by-3 tiled regions (9 images) at 20× magnification. Core biopsies were imaged with single 10× regions, and the TMA was imaged on the Zeiss AxioScan.Z1 platform using 5 × 5 tiled regions at 10× magnification. All tiled images were stitched in ZenBlue software. TIFFs with original signal were exported for analysis in Cell Profiler[63] software. The Cell Profiler pipeline included: DAPI smoothing using the Gaussian filter method, primary object identification from smoothed DAPI using the adaptive thresholding Otsu method on default settings, with clumped objects being distinguished and divided using the intensity setting. Primary object area, shape, and DAPI intensity were measured. Primary objects were then expanded a fixed pixel distance, and mean signal intensity for all other channels was measured in this expanded cellular region. Cells touching the image border were excluded from analysis. Spreadsheet outputs were then analyzed in FlowJo software (FlowJo LLC). Single cells were gated by nuclear area and shape, and single cell positivity for K19, K14, and VIM were determined by gating using tumors negative for each marker as controls. Gate shape was optimized to minimize false positivity from nonspecific channel bleed-through. Cell identity was then mapped onto X vs. Y location dot plots called "state maps", and these digital reconstitutions of cellular phenotypes in the tumor were consistent with visually called phenotypes in tumor images. Regions encompassing normal ductal structures, or in situ lesions, were identified with pathologist assistance and omitted from analysis.

**Image cytometry of patient-derived xenograft tumors**. Tissue microarray slides with 31 PDX and three normal breast tissue controls were provided by Mike Lewis at Baylor College of Medicine, Houston, TX. Spots were approximately 4 mm × 4 mm and arranged across three slides. Tumors are described in detail in a separate publication[33] and at www.bcxenograft.org, including molecular subtype and patient ER/PR/HER2 IHC status, mouse passage number for the tumors ranged from 4–8 passages. Slides were prepared and stained as detailed above (Image Cytometry of Primary Tumor Samples) with antibodies against K19 (1:300), K14 (1:300), VIM (1:200), and Ku80 (1:100). Secondary antibodies included the addition of goat-anti-rabbit-dylight755 marking Ku80 nuclei, all 1:200. Slides were imaged on the Zeiss AxioScan.Z1 platform, where circular scan areas were hand-drawn around all TMA spots and regions were imaged as 5 × 5 tiled regions at 10× magnification, stitched, and exported from ZenBlue software. Images were analyzed in CellProfiler including: DAPI smoothing, primary object identification from smoothed DAPI using the adaptive thresholding Otsu method on default settings, with clumped objects being distinguished using the intensity setting and divided using the propagation setting. Primary object area, shape, mean DAPI intensity, and mean Ku80 intensity were measured in this nuclear area. Primary objects were then expanded a fixed pixel distance, and mean signal intensity for all other channels was quantified in this expanded cellular region. Cells touching the image border were excluded from analysis. Spreadsheet outputs were then analyzed in FlowJo software. Single human tumor cells were gated based on positivity for Ku80, and single cell positivity for K19, K14, and VIM were determined by gating, using tumors negative for each marker as controls. TMA regions with compromised tissue fidelity were omitted from the analysis. HCC70 xenografts (Fig. 6) were imaged on the Zeiss Axio platform and analyzed using this method.

**Image cytometry of cancer cell lines**. Cancer cell lines were plated in appropriate media and allowed to adhere overnight, followed by various experimental treatments. At endpoint, cells were fixed by adding equal volume of 4% paraformaldehyde (Electron Microscopy Sciences) solution with 1 mM $MgCl_2$ (Sigma) to the well media. Wells were then washed with PBS and permeabilized in a 0.3% Triton-X100 solution (Thermo). Primary antibodies were diluted in PBS with 2% BSA and incubated overnight at 4 °C. Cells were subsequently washed with PBS-Tween and incubated 1 h at room temperature in a secondary labeling solution including 1 µg/ml DAPI and combinations of secondary antibodies against primary host species (1:300 in 2% BSA). Wells were then washed with PBS-Tween, filled halfway with PBS, and either imaged immediately or stored at 4 °C. Cell imaging for Figs. 1–4 was performed on the Olympus ScanR Platform at 10× magnification capturing four images per well in 384-well plates, and nine images-per-well in 96-well plates. Single-cell nuclear and cytoplasmic fluorescent intensities were calculated using the Olympus ScanR Analysis Software: the DAPI-positive region of each cell was used as a boundary to quantitate nuclear signal, and a 10 pixel annulus around the nucleus was used to quantitate cytoplasmic signal, omitting nuclear signal. Cells touching the border of the image were removed from analysis. Imaging for Figs. 2d, 5, and 6 was performed on the INCELL 6000 platform (GE Biosciences) using the GE INCELL Analyzer analysis software to calculate cytoplasmic and nuclear signal in identical methodology as described above. FlowJo analysis software was used to identify cell phenotype. Marker positivity was defined using marker-negative controls, including VIM/K14-negative luminal B cell lines, and K14/K19-negative claudin-low lines for cell line phenotyping (Fig. 1i). "High" expression was defined as mean-cell mean-fluorescent-intensities exceeding the in mean-cell MFI in DMSO wells. For image presentation in the figures, the same image brightness and contrast settings were applied across all experimental samples and conditions within an experiment.

**Cell line gene expression analysis**. Publicly available BCCL gene expression data[34] was queried for the expression of three sets of 25 genes including those preferentially expressed in luminal and myoepithelial cells, identified by sorting experiments in normal breast tissue[2,4–6,30,31,35], as well as 25 mesenchymal/EMT-transition genes[7,36]. Pearson coefficient clustering and heatmap generation were performed using GENE-E software (Broad Institute). Cell line subtype was determined through previously described 4-class (Luminal B, Basal-like, Claudin-low, HER2E) intrinsic subtyping[18].

**Calculating heterogeneity**. The Shannon diversity index is used as a metric of cell-state heterogeneity throughout this work. Cell state frequencies were calculated using flow cytometry software (FlowJo) as described above. For each tumor, PDX, or cell line, the proportion of each cell state ($P_i$) was calculated by dividing cell state number by the total cell number in the population. The Shannon diversity index ($H'$)[32] was then calculated by multiplying $P_i$ by the $\log_2$ of $P_i$, for each cell state, then summing these numbers for the total number of states (S).

$$H' = -\sum_{i=1}^{S} Pi \ln Pi$$

**Therapeutic screening**. Drug screening plates were designed and created as previously described[64]. Briefly, 96-well master plates with 7-point dilutions of 119 inhibitors at 10× concentration were plated into three, 384-well plates at 5 µl drug per well using the EP Motion automated dispensing system (Eppendorf). Control wells with equal volumes of DMSO (Sigma) were also included. 384-well plates are kept at −20 °C until use, at which point they were thawed for 1 h at 37 °C and spun down at 800g. Cells were plated directly into warm drug plates using EP Motion (Eppendorf) automated pipettor in 50 µl media. Plates were sealed with AeraSeal (Excel Scientific) and incubated for 72 h at 37 °C in a 5% $CO_2$ atmosphere. The CellTiter 96 cell viability kit (Promega) was used to measure cell viability, calculated as a percent of proliferation comparing well signal to a negative control (DMSO), after subtracting positive control signal (cell-free media). All drug screening data from Fig. 2 is present in Supplementary Data 2. Maximal inhibition (Emax) and the projected maximal inhibition (Einf) were both calculated using GraphPad Prism software (V5, GraphPad Software Inc.). The FDA-approved cytotoxic therapy screen (Supplementary Fig. 10b) was constructed as previously described[19], containing agents purchased from Cayman, Sigma and Selleckchem, and analyzed as described above. Follow up experiments, including all other dose–response curve generating experiments in this study, were performed in 384- or 96-well plates using the CellTiter 96 kit, with each condition run in triplicate wells, placed in distinct areas of the plates to normalize for edge effects. Combination indices (CI) were calculated from replicate, fixed-ratio, dose escalation experiments using the Chou and Talalay method[48] with Compusyn software (Combosyn). CI values were reported at 75% and 90% inhibitory values (CI75 and CI90, respectively).

**RNA sequencing**. Two RNAseq runs were analyzed in this manuscript and will be detailed separately. RNA sequencing data presented in Figs. 2–5 was obtained as follows: Total RNA was isolated with TRIzol (Invitrogen) from HCC1143 cells following a 6-day treatment with 0.05% DMSO (Sigma), 1 µM Trametinib, 1 µM BEZ235, or 1 µM of both agents in combination (1:1), with drug replenished at day 3. cDNA libraries were generated using the Agilent SureSelect Strand Specific RNA kit (Agilent) using 150 ng total RNA input and following the manufacturers protocol. cDNA Libraries were sequenced on the Illumina HiSeq 2000 using 50 bp single read reads, grouping eight samples per lane. Base calling was performed using Illumina RTA (v1.13.48) and conversion to FASTQ was performed using CASAVA (v1.8.2, Illumina). Reads were then trimmed to 44 bases, discarding the first 4 bases, the next 44 bases were kept. Trimmed reads were aligned to the hg19 genomes using Bowtie software (v1.0.0) allowing up to three mismatches and require best unique matches. Custom R scripts were used to count tags that aligned to the exons of UCSC RefSeq gene models to calculate RPKM values.

For the second RNAseq analysis presented in Fig. 6, HCC1143 cells were treated in triplicate with 1 µM Trametinib, 0.5 µM BEZ235, 2 µM JQ1, Trametinib + JQ1, BEZ235 + JQ1, or a DMSO control for 72 h. Total RNA was isolated using the QIAGEN RNeasy mini kit according to manufacturer instructions. RNA was run on the Bioanalyzer (Agilent) to verify integrity. cDNA libraries were constructed with the Illumina Trueseq Sample Prep Kit v2 according to manufacturer's instructions, using 150 ng of total RNA input. cDNA libraries were sequenced on the NextSeq500 using 75 bp single-end reads, grouping nine samples per lane. Base calling was performed using Illumina RTA (v2.4.11) and de-multiplexing and conversion to FASTQ were performed using Bcl2fastq (v2.17.1.14, Illumina). Reads were then trimmed to 44 bases, discarding the first 4 bases, the next 44 bases were kept. Trimmed reads were aligned to the hg19 genomes using Bowtie[65] software (v1.0.0) allowing up to three mismatches and require best unique matches. Custom R scripts were used to count tags that aligned to the exons of UCSC RefSeq gene models to calculate RPKM values. All RNAseq FASTQ and RPKM.txt files can be found on the GEO Omnibus under accession number GSE82032.

**Geneset generation**. All genesets used in this study are presented in Supplementary Data 3. Genesets present in the molecular signature database (MSigDB, Broad Institute) were taken as is and are available at www.broadinstitute.org/gsea/msigdb/collections.jsp, the original study that generated the geneset is cited when discussed in our work. Genesets were also generated from studies not included in the MsigDB. A collection of 32 breast phenotype-related genesets was compiled and included genesets from study examining overlap in gene expression from historical mammary gland population-sorting experiments[39], a study examining intrinsic subtypes of breast cancer[41], a study profiling the claudin-low subtype of breast cancer[40], a study examining classical myoepithelial markers[31], and a study examining the epigenetic determinants of the human breast[2]. A second compilation including 25 chromatin modifier enzyme activity-related genesets was compiled from MSigDB as well as a study examining BRD4 binding sites in basal-like BCCLs[50], a study examining gene expression changes following ectopic BRD4 expression in a mammary cancer line[49], and a study examining gene expression changes following ectopic expression or knockdown of EZH2[14]. A final set of 13 transcription factor activity-related genesets was compiled from MSigDB and studies examining ectopic FRA1 expression in BCCLs, ectopic GATA3 expression in MDAMB231 cells, ZEB1 overexpression and knockdown in lung cancer cell lines, and a study examining change with ectopic expression of SNAI1 in the transformed breast cell line MCF10A[55]. To reduce geneset size for optimal GSEA results, genesets were compiled of the most statistically significant upregulated or down-regulated genes, limiting size to 500 genes. Genesets generated from Affymetrix DNA microarray experiments used the NetAffx Query tool for Affymetrix gene ID conversion (Affymetrix.com).

**Geneset enrichment analyses**. Geneset enrichment analysis[57] was performed using GSEA software (Broad Institute). Ranked lists of differential gene expression (log2 Fold Change) between DMSO control cells and treated cells were created from RPKM RNAseq data. Pre-ranked GSEA was performed with 1000 permutations and default values for other parameters. Ranked lists were run against a compilation of 32 breast phenotype genesets, a compilation of 25 chromatin modifier activity-related genesets, and a compilation of 13 transcription factor activity-related genesets, the latter was run with the C6_All oncogenic signature database (www.broadinstitute.org/gsea/msigdb/collections.jsp) to obtain more data and to reduce false positivity related to testing small geneset groups. All GSEA results are available in Supplementary Data 3.

**DNA sequencing, SNV and indel calling, and copy number analysis**. Full details of the whole exome and WGS analysis is available in Supplemental Methods. Briefly, total DNA from HCC1143 cells following a 6-day treatment with 1 μM Trametinib, 1 μM BEZ235, or DMSO using the Qiagen DNAeasy Blood and Tissue kit (Qiagen). Whole-exome DNA sequencing libraries were prepared in triplicate from these gDNA extracts using KAPA Hyper-Prep Kit (KAPA Biosystems) with Agilent SureSelect XT Target Enrichment System and Human All Exon V5 capture baits (Agilent Technologies). Next-generation sequencing was carried out using the Illumina HiSeq 2500 platform by the OHSU Massively Parallel Sequencing Shared Resource (MPSSR) to an average depth of 100× per library replicate. FastQ data files were aligned and processed using BWA MEM (0.7.12, GATK, Broad Institute). Bam files for replicate libraries were merged and somatic variants were called using MuTect (1.1.4, GATK, Broad Institute) between samples in pairwise, all-against-all approach, as well as between treatments and previously generated whole-exome data for the HCC1143-BL cell line, described by and available from Daemen et al.[34], which was used as a "matched normal" for calling somatic variants in the above treatment libraries. Further insertion and deletion (indel) calling was done on the bam files for treatment samples using GATK4 MuTect2 (2.1-beta, Broad Institute) pairwise between treatments and against the HCC1143-BL line. Variant and indel calls were filtered by the presence in dbSNP database (https://www.ncbi.nlm.nih.gov/projects/SNP/), as well by frequency and depth, and were further scrutinized and hand-curated using various software tools (see Supplementary Methods). All raw exome-seq files can be found in the Sequence Read Archive (www.ncbi.nlm.nih.gov/sra) under the accession SRP125560 (WES) and VCF files are available in Supplementary Data 4. For WGS, the remaining aliquots of the HCC1143 pre-capture sequencing libraries were pooled by treatment type, and indexing and pre-sequencing library amplification PCR was carried out following Agilent SureSelect XT post-capture indexing protocol (omitting the capture and clean-up steps). The resulting WGS libraries were sequenced to an average depth of 0.5× across the entire genome on the Illumina NextSeq 500 by the OHSU MPSSR. FastQ data files containing paired-end sequencing reads (75 bp) were aligned and processed as described above and copy-number gain or loss was determined using ichorCNA software[66] (https://github.com/broadinstitute/ichorCNA/) and the panel of normals provided with the software. Resulting log-2 ratios across the entire genome were compared between treatments to identify possible differences in copy-number gain/loss. All raw WGS files can be found in the Sequence Read Archive (www.ncbi.nlm.nih.gov/sra) under the accession SRP144106 (LP-WGS) and segmentation files are available in Supplementary Data 4.

**Cell death assays**. Cells were plated and allowed to adhere overnight. The next morning cells were treated with inhibitors and incubated with 250 nM YOPRO1 dye (Thermo). Time course imaging was performed on the IncuCyte ZOOM Live Cell Imaging System (Essen Bioscience), taking phase images and green channel fluorescence images every 12 h throughout a 72 h treatment course. Green objects per image were quantified in the Incucyte software. Total cells per image were measured from phase images using a custom Cell Profiler pipeline. Briefly, this pipeline included: Color to gray image conversion, edge enhancement using the LoG method, image smoothing using a Gaussian filter setting, and primary object (cell) identification using the automatic threshold strategy and distinguishing and dividing clumped objects by intensity.

**Cell cycle analyses**. Cells were plated in six identical plates and allowed to adhere overnight. The next morning all plates were treated identically with inhibitors, and one plate was incubated with EdU (Sigma) for 12 h at a concentration of 10 μM. At the 12-hour mark, this plate was fixed, and EdU was added to a second plate. Every 12 h a subsequent plate was fixed and another was pulsed with EdU to capture the next timepoint. Cell fixation and permeabilization were carried out as described above (Image Cytometry of Cancer Cell Lines). Wells were then treated with a reaction buffer containing 2 mM CuSO₄ (Sigma), 8 μM AlexaFluor Azide 647 (LifeTech), and 100 mM sodium ascorbate (Sigma) in PBS and incubated 1 h. After washing with PBS, cells were stained for K19, K14, and VIM, imaged, and analyzed as outlined above (Image Cytometry of Cancer Cell Lines) using nuclear detection of 647-channel signal to quantify cellular EdU levels.

**Mass cytometry**. HCC1143 cells were treated with 1 μM BEZ235, 1 μM Trametinib, or with 0.05% DMSO and incubated at 37 °C for 72 h. Cells were then washed twice with PBS, trypsinized, and stained with 20 μM Cisplatin (Selleckchem) for 1 min at room temperature. Cells were fixed in 1.6% Paraformaldehyde (Electron Microscopy Sciences), followed by permeabilization with a 0.3% Saponin buffer (Sigma). Cytokeratin14, Cytokeratin17, and Claudin4 antibodies were conjugated to lanthanide metals using the MAX PAR conjugation kit (Fluidigm/DVS Sciences). $2 \times 10^6$ cells from each sample were incubated with the antibody panel overnight at 4 °C in Saponin buffer. Cells were stained in 1:1000 Natural Iridium (Fluidigm/DVS Sciences) in PFA buffer (1.6% PFA, 0.24% Saponin, PBS) for 20 min at room temperature. Approximately 500,000 cells were loaded into the CyTOF mass cytometer and data was acquired for 5 min. Cell events in the range of 10–75 pushes were recorded in the FCS file, with an average of 100,000 cells recorded for each sample. FCS data files were analyzed in Cytobank (Fluidigm/DVS Sciences). Gates for high expression were defined by mean-cell marker signal in DMSO wells.

**Computational modeling**. Models of drug-induced cell-state dynamics of HCC1143 cells under two distinct hypotheses (Darwinian selection, transition-mediated) were identified using measurements from 15 replicate wells taken every 12 h over a 72-hour horizon (Fig. 3a, b, d). These models assumed that each cell at each instant could be either dead or alive, and if alive, expressed either K14hi or K14low. Under each hypothesis, the number of K14hi live cells, K14low live cells, and dead cells over time under drug treatment were estimated based on state-selective death criteria. For example, for the K14hi Darwinian selection scenario, the number of K14hi live cells under Trametinib was set to the measured K14hi cell count, whereas the number of K14low live cells under Trametinib was set to the measured K14low cell count minus the estimated dead cell count (Supplementary Methods). The number of dead cells was estimated by multiplying measurements of cell death proportion and population total (Supplementary Methods). Under both hypotheses, the quantities of K14hi and K14low live cells over time under DMSO were estimated by distributing death equally between the measured K14hi and K14low cell counts (Supplementary Methods). The time course training data for each hypothesis-agent pair (e.g., K14hi Darwinian selection and Trametinib) were fed into a constrained $l_2$-regularized least-squares program with alternating minimization (as some measurements were illegible) to learn locally optimal dynamics (CVX optimization package[67]). Linear time-invariance was assumed because additional complexity was expected to overfit the data (Supplementary Methods). Model parameters are time-averaged rates of division, death, and transition of K14hi and K14low live cells; refer to Chapman et al.[68] for details. In the optimization program, these parameters were constrained according to hypothesis-specific assumptions on cell-state transition and death. Under the K14hi Darwinian selection scenario, for example, Trametinib-induced rates of cell-state transition and K14hi death were set to zero (Supplementary Methods). Evolution of HCC1143 cell populations following drug treatment (or DMSO) was simulated by propagating the hypothesis-specific models forward in time from appropriate initial conditions. The initial condition for each drug treatment simulation was the number of K14hi live cells, K14low live cells, and dead cells in each drug-treated well estimated at time zero. The initial condition for the baseline simulation was the average of such numbers over all DMSO wells. Change in subpopulation proportion (vs. DMSO) was computed from the simulated evolution of HCC1143 cell populations over time (Fig. 3h, Supplementary Fig. 18a, Supplementary Methods). MATLAB software (MathWorks) was used for all computational modeling.

**VIPER analysis**. The gene expression (raw counts) was normalized by the library size (total number of reads mapped to transcripts) and transformed to stabilize the variance by fitting the dispersion using a negative-binomial distribution as implemented in the DESeq package from Bioconductor[69]. To reduce the impact of systematic variability, affecting mainly low expressed genes, we focus our analysis only on mid-to-high expressed genes. The threshold to identify low expressed genes was defined by the data as follows: we fit a mixture of two Gaussian models to the probability density of expression, and used them to compute the likelihood ratio (LR) of high expression. Low expressed genes (LR < 1) were not considered for further analysis. While this procedure trimmed the expression profile to 10,313 genes, it should not affect the quality of VIPER results, as we have previously shown the analysis is strongly robust to partial signature representation[46]. Gene expression signatures were computed by comparing each perturbed sample vs. DMSO vehicle control. The VIPER algorithm, available from Bioconductor (http://bioconductor.org/packages/release/bioc/html/viper.html) was then used to estimate the relative activity of 5087 regulatory proteins, including transcription factors and signaling proteins. This analysis was based on a transcriptional regulatory model assembled by the ARACNe algorithm[70] from 1047 breast carcinoma tumors profiled by The Cancer Genome Atlas (TCGA). The regulatory model is available from figshare (https://doi.org/10.6084/m9.figshare.2750698). Top upregulated and down-regulated VIPER hits were analyzed with DAVID pathway ontology analysis against the KEGG, BIOCARTA, REACTOME, and PANTHER databases. All VIPER results are present in Supplementary Data 5.

**Single cell ATAC-seq library construction and analysis**. Single cell combinatorial indexing ATAC-seq libraries were prepared following the protocol described in Cusanovich et al.[52] using flow sorting at each stage of nuclei partitioning as opposed to dilution. All treatments were multiplexed in the same transposase-based indexing plate with the wells of the plate corresponding to each treatment condition. Indexing at this level has the distinct advantage of preventing any potential cell collisions from occurring between cells from two separate conditions as the treatment condition is encoded in the transposase barcode prior to pooling and redistribution. After combinatorial indexing library construction, all PCR wells were pooled and assessed on a Bioanalyzer High Sensitivity DNA chip (Agilent) prior to sequencing on an Illumina NextSeq 500 according to protocols outlined in Amini et al.[71]. Sequence reads were demultiplexed using SCI-seq software provided in Vitak et al.[72] prior to alignment using bowtie2[65]. PCR read duplicates were removed on a cell-level basis, again using SCI-seq software. Index combinations were then filtered to exclude background reads and to only retain those containing at least 1000 uniquely aligned sequence reads with a mapping quality of at least 10. The combined alignment file was then used for peak calling using MACS2[73] with default parameters. Reads and peaks were then used to construct a counts matrix as described in Cusanovich et al.[52] and filtered to retain only cells with at least 1000 on-target reads, and sites that contain reads from at least 50 cells which was then used to perform latent semantic indexing (LSI), retaining dimensions 1 through 15. On the LSI matrix we then carried out t-SNE[74]. To compute transcription factor deviation scores we used chromVAR[53] using the transcription factor motif collection provided by the tool and plotted deviation Z-scores on the respective LSI t-SNE visualizations.

**Animal studies**. Mice were handled in accordance with the OHSU Institutional Animal Care and Use Committee (IACUC). A total of $2 \times 10^6$ HCC70 cells in 50% Matrigel (Corning) + 50% RPMI1640 were bilaterally injected into the left and right fourth mammary glands of 4–6-week-old nonobese-diabetic (NOD)/SCID/$\gamma$-chain null (NSG) mice. Tumors were allowed to grow until they reached 100 mm³, at which point all mice were randomized into treatment groups of four mice per group, with a total of eight tumors per group. Based on other xenograft studies this was sufficient to detect mean differences in tumor size between groups greater than 1 standard deviation at 5% significance (using two-tailed Student's T-test) with 95% power. Following randomization, mice were treated once daily, for 21 days in the following groups: (1) oral gavage (OG) control vehicle (10% 1-methyl-2-pyrrolidone/90% PEG300) and intraperitoneal injection (IP) vehicle (10% DMSO in 2-hydroxypropyl-B-cyclodextrin, 10% w-v in water); (2) OG control vehicle + 50 mg/kg JQ1 IP; (3) IP vehicle + 40 mg/kg NVP-BEZ235 by OG; or (4) a combination of 50 mg/kg JQ1 IP and 40 mg/kg NVP-BEZ235 by OG. Caliper measurements and tumor volume calculations were performed every 2–3 days using the $V = (L \times W^2)/2$ equation. Mice were euthanized following the 21-day treatment period according to IACUC protocol and tumors were harvested, formalin fixed, paraffin embedded, sectioned and immunofluorescently interrogated (see image cytometry of PDX tumors). Investigators were not blinded when assessing tumor volumes, but were blinded during immunofluorescent analysis.

**Statistical analysis**. All statistical analyses were performed using GraphPad Prism software. Data is presented as mean with standard error when showing averages across biological replicates, or mean with standard deviation within representative experiments, all experiments were repeated at least three independent times. The number of replicates was chosen based on prior knowledge of specific experimental variability. For determining significance, replicate data was first tested with the D'Agostino & Pearson omnibus normality test. Normally distributed data was compared with the two-tailed Student's T-test, or paired Student's T-test for paired data. Non-normally distributed data was compared using the Mann–Whitney test, or the Wilcoxon matched-pairs signed rank test for paired data.

**Data availability**. All analyzed tumor images (Fig. 1, Supplementary Fig. 1) and associated image cytometry plots are shown in Supplementary Data 1. RNA sequencing data are available at the GEO data repository with the accession code GSE82032. Drug screen data (Fig. 2) is provided in Supplementary Data 2. Genesets and GSEA results (Figs. 2, 4–6, Supplementary Fig. 5 and 14) are provided in Supplementary Data 3. All raw DNA sequencing files and analyses (Supplementary Fig. 7) can be found under the following Sequence Read Archive accessions (www.ncbi.nlm.nih.gov/sra): SRP125560 for whole exome-seq files, SRP144106 for low-pass WGS files. Vcf files for all MuTect analyses (Supplementary Table 1 and 2) are included in Supplementary Data 4 along with .seg files from copy number analysis (Supplementary Fig. 7). VIPER analysis results and pathway ontology results (Fig. 4, Supplementary Fig. 9) are presented in Supplementary Data 5. Raw sciATAC-seq motif enrichment results (Fig. 6, Supplementary Fig. 13) are presented in Supplementary Data 6. All cell profiler image analysis pipelines, tumor images, and source data are available upon request.

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

## Acknowledgements

We would like to thank Dr. Megan Troxell of the OHSU Pathology Department for her assistance in obtaining and analyzing primary breast cancer tumor samples from the OHSU Biobank, the Dr. Jay Bradner Lab of the Dana-Farber Cancer Institute for their generous donation of JQ1 for our animal studies, Dr. Dimitri Rozanov for providing cytotoxic therapy screening plates, Dr. Dan Georgess for his assistance with statistics, Elmar Bucher for the generation of data management Python code, and Dr. Jason M. Link for his generation of an image analysis pipeline. We acknowledge Dr. Bob Searles and MPSSR core at OHSU for their sequencing efforts. T.R. was supported by the Ruth L. Kirschstein PMCB T32 Training Grant 5T32GM071338-09, Vertex Pharmaceuticals Scholarship, and Tartar Trust Fellowship. E.M.L. was supported by an ACS postdoctoral fellowship 118795-PF-10-022-01-CSM, M.P.C. was supported by the NSF Graduate Research Fellowship Program and the Berkeley Fellowship for Graduate Study. A.C was supported by the National Cancer Institute (NCI) Outstanding Investigator Award (R35 CA197745), the NCI Research Centers for Cancer Systems Biology Consortium (U54 CA209997), and two National Institutes of Health (NIH) instrumentation grants (S10 OD012351 and S10 OD021764). C.J.T. is supported by the NIH Center "Systems Biology

of Collective Cells Decisions" through Stanford University NIH P50GM107615. R.C.S. is supported by the National Institutes of Health, National Cancer Institute R01 CA129040 and R01 CA100855, the Department of Defense Breast Cancer Research Program BC103625, the Colson Family Foundation, and the Prospect Creek Foundation. R.C.S. and J.W.G. are supported by the NCI U54 grant CA209988. J.W.G. is supported by the National Institutes of Health, National Cancer Institute grant U54 CA112970, by the Susan G. Komen Foundation, and the Prospect Creek Foundation. Sequencing and Flow Cytometry work was performed in OHSU Shared Resources supported by the Knight Cancer Institute Cancer Center Support Grant 5 P30 CA69533.

## Author contributions

T.R., E.M.L., J.R., J.W.G., and R.C.S. designed the study. T.R., E.M.L., R.C.S., and J.W.G. wrote the manuscript. M.L. and L.D. generated the patient-derived xenografts and T.R. imaged the tumors. T.R. and K.C. performed primary tumor immunofluorescence. T.R. performed cell line gene expression analysis, cell culture (with E.M.L.), therapeutic screening (with J.R.), image cytometry, cell death and S-phase quantitation assays, geneset enrichment analysis, and cell line synergy testing. K.J.C. and P.S. performed the mass cytometry experiments and analyses. C.B. and P.S. performed the whole exome sequencing, LP-WGS sequencing, and mutation analyses. M.P.C. and C.J.T. developed the computational models with statistical help from A.J.A. T.R. and N.W. performed the RNA sequencing and C.P. performed the alignment and expression analysis and other bioinformatics support. M.A. and A.C. performed VIPER analyses. T.R. and N.K. carried out the animal studies. T.R. and A.F. performed sciATAC-seq assays. A.A. and T.R. analyzed sciATAC-seq data.

## Additional information

**Competing interests:** Andrea Califano is founder, equity holder, consultant, and director of DarwinHealth Inc., a company that has licensed the VIPER algorithm used in this manuscript from Columbia University. Columbia University is also an equity holder in DarwinHealth Inc. Mariano J. Alvarez is an equity holder and consultant of DarwinHealth Inc.

