## [Peer Review File · Nature Communications]

Reviewers' comments:

Reviewer #2 (Remarks to the Author):

In the current revised manuscript, Risom et al. have added multiple experiments and addressed satisfactorily several comments. In particular, the use of digital image analysis has improved the manuscript substantially and provided an objective and unbiased assessment of the phenotypic heterogeneity in tumors, patient-derived xenografts, and cell-lines. In addition, experiments have been repeated in multiple cell lines, demonstrating consistent results and suggesting some generalizability of their findings.

However, the authors failed to provide direct genetic evidence to rule out the hypothesis that resistance to targeted therapies (i.e., anti-PI3K and anti-MEK) is caused by clonal selection. The authors have indeed compiled significant evidence to suggest that cell-state transition rather than clonal selection underpins the phenotypic responses to MEK and PI3K/mTOR inhibitors. Although their conclusion may be supported by the alternative explanation of alterations in chromatin regulation, the evidence provided is circumstantial.

The added computational model may provide additional *in silico* evidence to support the author's conclusion; however, it appears that the computational model was developed and trained (Chapman et al.) with their own experimental results. No cross-validation nor testing in independent datasets has been described. Therefore, data over-fitting cannot be ruled out, and the validity of the computational model is questionable.

Moreover, the fact that the cancer cells return to a phenotypically heterogeneous state once treatment withdrawn does not completely rule out clonal selection. Studies on lung cancer and gastrointestinal stromal tumors (GISTs) cell lines have demonstrated that small amounts of residual cells lacking the secondary mutations, such as the T790M EGFR mutation in lung cancer, are found within resistant mutant cells despite a generally homogeneous cell population upon treatment, and that once treatment is withdrawn these cells proliferate and more overt heterogeneity is observed again.

Therefore, the authors ought to provide direct evidence that the so-called DTP cells do not differ genetically from the cells before treatment. Ideal evidence that phenotypic heterogeneity is not associated with genetic heterogeneity would be accrued using single cell sequencing of tumor cells following selection of cells based on phenotypic features investigated in this study. That may not be readily available and beyond the scope of this study. However, high-depth (>500x) whole-exome sequencing analysis of pre- and post-treatment cell lines is feasible and would be satisfactory.

Minor:

1) Figure Legend 3, "(f)" is repeated. Second should read "(h)".

Reviewer #3 (Remarks to the Author):

In their revised manuscript "Managing drug-induced differentiation-state plasticity in basal-like breast cancer to improve therapeutic control", Risom et al. present a substantially expanded and

more complete study characterizing the phenomenon of drug induced differentiation state changes in breast cancer. By adding a substantial amount of new experiments and presenting their data in a more consistent and complete way, the authors successfully address all major concerns of the reviewers.

One major concern of all reviewers was the limited number and inconsistent use of primary samples and breast cancer cell lines. To address this point, the authors have now included a larger panel of cell lines as well as additional 31 patient-derived xenograft samples, which is impressive. In addition, the authors have substantially expanded the characterization of cellular states in breast cancer samples of different subtypes, and provide comprehensive image cytometry for all primary samples, patient derived xenografts and cell lines. The phenotypic screen using 119 compounds has been expanded to a second cell line and the entire primary data is now included as supplementary information.

While it previously remained unclear whether the observed drug induced state changes were pre-existing or a consequence of drug treatment, the authors now show (based on EdU incorporation and sophisticated mathematic models) that the emergence of drug tolerant persister cells is more likely explained by adaptive changes of cellular states rather than Darwinian selection. In support of this model, the authors have also added comprehensive single-cell ATAC-seq analyses of cells treated with MEKi and PI3Ki, which also addresses previous concerns about the use of just a few IF markers. Beyond changes in single markers, results from single-cell ATAC-seq demonstrate that the chromatin architecture of MEKi/PI3Ki treated cells undergoes complex changes that are associated with certain chromatin modifier and transcription factor networks. As key regulator of this cellular plasticity, they (now more systematically) identify BRD4 and demonstrate that the BET inhibitors can be used to interfere with this phenomenon. Importantly, ATAC-seq analyses reveals that combined treatment with JQ1 prevents chromatin changes upon PI3K inhibition, which establishes a mechanistic model explaining the synergy between both agents.

In summary, Risom et al. have undertaken a major revision of their manuscript and added an impressive amount of new data and analyses. While neither the concept of epigenetic adaptation to drugs nor the synergy between PI3K and BET inhibitors are completely novel, the revised manuscript characterizes epigenetic drug-response mechanism in basal-like breast cancer at an unprecedented level of depth, and establishes a mechanistic rationale for combination therapies that are urgently needed in the clinic. For ongoing debates about the origin of resistance, the manuscript provides an impressive example of truly adaptive epigenetic resistance. Furthermore, comprehensive datasets of this study that are now provided in a very organized and complete way, will serve as a useful resource for other studies focused on targeted therapies in breast cancer.

While all my major concerns are fully addressed, I found a few points that the authors could consider to further improve the clarity of the manuscript.

1. To address shared concerns about the small number of used cell lines and primary samples, the authors have characterized heterogeneity now in a large set of cell lines and primary patient samples. While I applaud this effort, the presentation of primary data in Figure 1 and 11 (!) Supplemental Figures is too extensive and distracting from the main messages of the paper. I strongly recommend condensing and simplifying the presentation of these results, both in the main figure and in the supplement. It would be sufficient and much clearer to reduce the presentation of primary analyses to just a few examples (to help understanding about how the data was obtained), and then present the larger panel of analyzed samples in an integrated way (like in Fig. 1b, e, j).

2. The authors should re-consider the presentation of synergistic effects in Figure 5b. When I first reviewed the manuscript, I found the effect scale very disappointing, but now understand that single agents first trigger a massive clearance of bulk cells until a plateau is reached, which can be

completely prevented through combined therapies. While impressive, Figure 5b does not illustrate this effect in a clear way, and I am concerned that many readers will be rather disappointed with what's shown. As an alternative, the effects could be shown as simple proliferation curves (cell counts) at one selected dose with or without showing different subpopulations (data might already be available from microscopy analysis). The authors could also consider sequential treatment, where they first eliminate single drug responders and then follow effect +/-JQ1 addition. There are many ways, but I definitely would consider these options to avoid confusion.

3. I would reconsider the title. The current version seems complicated, and the terms "Managing" and "therapeutic control" sounds a bit unscientific. I would focus more on the plasticity of differentiation states as a major driver of resistance...and if its getting too complex omit mentioning the therapeutic angle (which is still early days).

These are just suggestions and, from my perspective, should not be viewed as conditional for a positive evaluation of this manuscript. I congratulate the authors to a very impressive and successful revision.

We thank you for the review of our updated manuscript that was titled “Managing drug-induced differentiation-state plasticity in basal-like breast cancer to improve therapeutic control”. We appreciate the comments from the reviewers and have implemented all suggested changes, including the suggested updates to figure presentation and manuscript text to improve the manuscript clarity. Further, we have added the whole exome sequencing experiment as requested by Reviewer 2 to further support and strengthen our conclusions. All changes to the manuscript text are highlighted as requested. Our response to all major reviewer concerns is listed point-by-point below.

Reviewer 2:

In the current revised manuscript, Risom et al. have added multiple experiments and addressed satisfactorily several comments. In particular, the use of digital image analysis has improved the manuscript substantially and provided an objective and unbiased assessment of the phenotypic heterogeneity in tumors, patient-derived xenografts, and cell-lines. In addition, experiments have been repeated in multiple cell lines, demonstrating consistent results and suggesting some generalizability of their findings.

However, the authors failed to provide direct genetic evidence to rule out the hypothesis that resistance to targeted therapies (i.e., anti-PI3K and anti-MEK) is caused by clonal selection. The authors have indeed compiled significant evidence to suggest that cell-state transition rather than clonal selection underpins the phenotypic responses to MEK and PI3K/mTOR inhibitors. Although their conclusion may be supported by the alternative explanation of alterations in chromatin regulation, the evidence provided is circumstantial.

As detailed below, we have conducted WES as requested.

The added computational model may provide additional in silico evidence to support the author’s conclusion; however, it appears that the computational model was developed and trained (Chapman et al.) with their own experimental results. No cross-validation nor testing in independent datasets has been described. Therefore, data over-fitting cannot be ruled out, and the validity of the computational model is questionable.

We appreciate this point. The model presented in the paper was indeed trained using the timecourse data presented in Fig. 3a. The purpose of this computational modeling was to test potential mechanisms of generating DTP states, and to achieve this, the resulting model was not tested on the data used to train it. Rather, we assessed whether certain cell functions (e.g. cell switching, selective cell death) were necessary to predict the trends of the experimental data. To do this, we altered model parameters (e.g. whether cells could switch states, which cells could die) and compared the predictions of the model with those altered parameters to the experimental data. When viewed in the context of the empirical findings (e.g., Fig. 3e,f, where apoptosis is blocked experimentally with ZVAD), the modeling supports the laboratory

experiments in evaluation of which cell functions could underlie the observed differentiation-state enrichment. Together, the *in silico* model and empirical results throughout the paper both argue against Darwinian selection of genomic clones and for epigenomic state-change as the stronger driver of the differentiation-state enrichment.

Moreover, the fact that the cancer cells return to a phenotypically heterogeneous state once treatment withdrawn does not completely rule out clonal selection. Studies on lung cancer and gastrointestinal stromal tumors (GISTs) cell lines have demonstrated that small amounts of residual cells lacking the secondary mutations, such as the T790M EGFR mutation in lung cancer, are found within resistant mutant cells despite a generally homogeneous cell population upon treatment, and that once treatment is withdrawn these cells proliferate and more overt heterogeneity is observed again.

Therefore, the authors ought to provide direct evidence that the so-called DTP cells do not differ genetically from the cells before treatment. Ideal evidence that phenotypic heterogeneity is not associated with genetic heterogeneity would be accrued using single cell sequencing of tumor cells following selection of cells based on phenotypic features investigated in this study. That may not be readily available and beyond the scope of this study. However, high-depth (>500x) whole-exome sequencing analysis of pre- and post-treatment cell lines is feasible and would be satisfactory.

We agree with the reviewer that genomic sequencing could directly test the presence of Darwinian clonal selection in the cell line following Trametinib and BEZ235 treatment. We have now performed these experiments and carried out next-generation sequencing on whole-exome capture libraries to look for the presence of genomic changes in the HCC1143 cell line following Trametinib and BEZ235 treatment, with a DMSO-treated population as a control. The libraries were prepared in triplicate, to a combined average depth of 300x (cost and time constraints prevented achieving our target of 500x). Importantly, no variants were called between the three treatments (DMSO, Trametinib and BEZ235) throughout the 85% of the exonic regions having coverage greater than 100x depth. This suggests that genomic selection of cells with distinct mutational profiles did not occur during treatment, arguing against a mechanism of Darwinian selection.

In areas of the genome that had relatively poor sequencing coverage (30-60x depth), some variants were called between experimental groups at our threshold variant allele frequency (~5-10%) using MuTect software. Upon further examination, however, these variants were inconsistent across sample replicates, inconsistent when the samples were compared to different groups (e.g. variants found in Trametinib vs BEZ235 were not seen in Trametinib vs. DMSO), and often mapped to the same start and stop position of the reference sequence. Therefore, we attribute these variants to experimental noise arising from polymerase errors propagated in library preparation steps and/or sequencing errors, and not as evidence of genomic selection. Additionally, since the variants observed in the low-coverage regions showed a <10% variant allele frequency, they still could not support the hypothesis of Darwinian selection to generate a >40% change in K14-hi expressing cells following treatment (see Fig. 3e).

Finally, to assess further the possibility of clonal selection, we compared the variant profiles of Trametinib, BEZ235, and DMSO-treated HCC1143 cells to patient-matched normal cells (HCC1143-BL). This analysis identified variants in HCC1143 when compared to normal cells - these variants were present in all three treatment conditions at similar frequencies (Figure A, below). The difference in allele frequency observed (<10%) of the major variants found in all

three HCC1143 conditions does not support the hypothesis of treatment-driven clonal selection. We would expect the variant allele frequency (VAF) to change by orders of magnitude if those variants were representative of subclones that were selected for or against on the basis of fitness. Although we don't have the statistical power to say that these 1-10% changes in VAF between treatments are not real, in our experience such fluctuations are well within the observed variability between multiple sequencing runs of libraries derived from the same samples.

Figure A. Sequencing variants between HCC1143 cells and their patient-matched normal B-lymphoblastoid sample (HCC1143-BL) are arranged along the X-axis with the chromosomal location displayed. Variant allele fraction normalized to ploidy is shown on the Y-axis, different color bars represent the normalized variant allele fraction found in DMSO (gray), BEZ235 treated cells (pink), or Trametinib treated cells (Cyan).

Taken together, these results argue against a phenomenon of Darwinian selection following Trametinib and BEZ235.

Minor:

1) Figure Legend 3, “(f)” is repeated. Second should read “(h)”.

Thank you for catching this. It is now corrected.

Reviewer 3:

In their revised manuscript “Managing drug-induced differentiation-state plasticity in basal-like breast cancer to improve therapeutic control”, Risom et al. present a substantially expanded and more complete study characterizing the phenomenon of drug induced differentiation state changes in breast cancer. By adding a substantial amount of new experiments and presenting their data in a more consistent and complete way, the authors successfully address all major concerns of the reviewers.

One major concern of all reviewers was the limited number and inconsistent use of primary samples and breast cancer cell lines. To address this point, the authors have now included a larger panel of cell lines as well as additional 31 patient-derived xenograft samples, which is impressive. In addition, the authors have substantially expanded the characterization of cellular states in breast cancer samples of different subtypes, and provide comprehensive image cytometry for all primary samples, patient derived xenografts and cell lines. The phenotypic screen using 119 compounds has been expanded to a second cell line and the entire primary data is now included as supplementary information.

While it previously remained unclear whether the observed drug induced state changes were pre-existing or a consequence of drug treatment, the authors now show (based on EdU incorporation and sophisticated mathematic models) that the emergence of drug tolerant persister cells is more likely explained by adaptive changes of cellular states rather than Darwinian selection. In support of this model, the authors have also added comprehensive single-cell ATAC-seq analyses of cells treated with MEKi and PI3Ki, which also addresses previous concerns about the use of just a few IF markers. Beyond changes in single markers, results from single-cell ATAC-seq demonstrate that the chromatin architecture of MEKi/PI3Ki treated cells undergoes complex changes that are associated with certain chromatin modifier and transcription factor networks. As key regulator of this cellular plasticity, they (now more systematically) identify BRD4 and demonstrate that the BET inhibitors can be used to interfere with this phenomenon. Importantly, ATAC-seq analyses reveals that combined treatment with JQ1 prevents chromatin changes upon PI3K inhibition, which establishes a mechanistic model explaining the synergy between both agents.

In summary, Risom et al. have undertaken a major revision of their manuscript and added an impressive amount of new data and analyses. While neither the concept of epigenetic adaptation to drugs nor the synergy between PI3K and BET inhibitors are completely novel, the revised manuscript characterizes epigenetic drug-response mechanism in basal-like breast cancer at an unprecedented level of depth, and establishes a mechanistic rationale for combination therapies that are urgently needed in the clinic. For ongoing debates about the origin of resistance, the manuscript provides an impressive example of truly adaptive epigenetic resistance. Furthermore, comprehensive datasets of this study that are now provided in a very organized and complete way, will serve as a useful resource for other studies focused on targeted therapies in breast cancer.

While all my major concerns are fully addressed, I found a few points that the authors could consider to further improve the clarity of the manuscript.

1. To address shared concerns about the small number of used cell lines and primary samples, the authors have characterized heterogeneity now in a large set of cell lines and primary patient samples. While I applaud this effort, the presentation of primary data in Figure 1 and 11 (!) Supplemental Figures is too extensive and distracting from the main messages of the paper. I strongly recommend condensing and simplifying the presentation of these results, both in the main figure and in the supplement. It would be sufficient and much clearer to reduce the presentation of primary analyses to just a few examples (to help understanding about how the data was obtained), and then present the larger panel of analyzed samples in an integrated way (like in Fig. 1b, e, j).

We agree. We have reduced the complexity of the results presented in Fig. 1a and in the Supplementary Figures that support Figure 1. We now show representative examples of image quantitation in new Supplementary Figures 1b, e as suggested by the reviewer, and have moved the bulk of the tumor images and image cytometry to a new Supplementary Data file (Supplementary Data). This simplifies the clarifies the explanation of how the analysis was performed, while still allowing access to all of the data if desired.

2. The authors should re-consider the presentation of synergistic effects in Figure 5b. When I first reviewed the manuscript, I found the effect scale very disappointing, but now understand that single agents first trigger a massive clearance of bulk cells until a plateau is reached, which can be completely prevented through combined therapies. While impressive, Figure 5b does not illustrate this effect in a clear way, and I am concerned that many readers will be rather

disappointed with what's shown. As an alternative, the effects could be shown as simple proliferation curves (cell counts) at one selected dose with or without showing different subpopulations (data might already be available from microscopy analysis). The authors could also consider sequential treatment, where they first eliminate single drug responders and then follow effect +/-JQ1 addition. There are many ways, but I definitely would consider these options to avoid confusion.

We thank the reviewer for this comment and agree that the graphs presented in Fig. 5 in the previous version did not focus the reader's attention on the combinatorial benefits of JQ1. We have added a new figure panel, Fig. 5c, which highlights the significant gains in maximal kill with the JQ1 + BEZ235 combinations, compared to BEZ235 alone. We feel this better presents the combinatorial effects to the reader, demonstrating that BEZ235 never reaches 100% kill, but this maximal death can be achieved with the addition of the BET inhibitor. Similar plots have been generated for BEZ235 + JQ1 examination in Luminal B lines (Supplementary Fig. 9b), as well as with Trametinib + JQ1 combinations (Supplementary Fig. 9d).

3. I would reconsider the title. The current version seems complicated, and the terms "Managing" and "therapeutic control" sounds a bit unscientific. I would focus more on the plasticity of differentiation states as a major driver of resistance...and if its getting too complex omit mentioning the therapeutic angle (which is still early days).

We appreciate this suggestion. We have arrived at a new title that we think succinctly captures the most impactful findings in our study; specifically that differentiation state plasticity is a resistance mechanism in basal-like breast cancer that can be targeted therapeutically. Our new title is:

"Differentiation-state plasticity is a targetable resistance mechanism in basal-like breast cancer".

These are just suggestions and, from my perspective, should not be viewed as conditional for a positive evaluation of this manuscript. I congratulate the authors to a very impressive and successful revision.

We would like to again thank the editor and reviewers for all their time and effort in reviewing our manuscript and believe that it has benefited substantially from this important input and is now more clear and impactful.

With kind regards,

Rosalie Sears and Joe Gray

Reviewers' comments:

Reviewer #2 (Remarks to the Author):

In this re-revised manuscript, the authors have provided some genetic evidence to rule out the hypothesis that resistance to targeted therapies (i.e., anti-PI3K and anti-MEK) is caused by clonal selection. They have performed a reasonably high-depth (300x) WES on one cell line (HCC1143) without and after treatment with Trametinib and BEZ235. Somatic mutation analysis was performed in pairwise, all-against-all approach, between treatments and using a matched lymphoblastoid cell line as a matched normal. The authors have concluded that there were no significant differences in the mutations profiles of the cells before and after each one of the treatments, supporting their conclusion that Darwinian selection did not occur after treatment.

Importantly, however, the authors did not include these crucial results in the re-revised manuscript although the methods were included; and only part of the data were presented in a rebuttal letter-only figure (Figure A). In its current form, it is rather difficult to critically assess the WES performed, and potential readers of the manuscript would not be able to visualize and assess these data. For instance, according to the rebuttal letter, about 15% of the genome was sequenced at lower depth (<100x) than most of the genome. Differences in mutations affecting these low-coverage areas were observed, but disregarded by the authors as likely sequencing errors. Which were these variants? Did they affect cancer genes? The authors ought to report on these crucial findings in the manuscript, and illustrate the data completely, either in a supplementary figure, or in the form of a vcf file listing the detected mutations in the different samples according to the different computational approaches. Furthermore, given that the treatment experiments of HCC1143 with Trametinib and BEZ235 were likely repeated, the authors have not provided evidence that the cell lines subjected to WES did show similar phenotypic shifts previously observed. Therefore, it is unclear whether the DNA sequenced matched the different phenotypes observed in the authors' previous experiments.

Taken together, the authors ought to describe and present in more detail the results of the WES analysis in the manuscript.

Dear Reviewer,

We thank you for your comments on our manuscript “Differentiation-state plasticity is a targetable resistance mechanism in basal-like breast cancer”. We agree with your points and have updated the manuscript to include new supplementary figure panels and a new supplementary table that describes the data and findings of the WES experiment, in line with your suggestions:

Importantly, however, the authors did not include these crucial results in the re-revised manuscript although the methods were included; and only part of the data were presented in a rebuttal letter-only figure (Figure A). In its current form, it is rather difficult to critically assess the WES performed, and potential readers of the manuscript would not be able to visualize and assess these data. For instance, according to the rebuttal letter, about 15% of the genome was sequenced at lower depth (<100x) than most of the genome. Differences in mutations affecting these low-coverage areas were observed, but disregarded by the authors as likely sequencing errors. Which were these variants? Did they affect cancer genes? The authors ought to report on these crucial findings in the manuscript, and illustrate the data completely, either in a supplementary figure, or in the form of a vcf file listing the detected mutations in the different samples according to the different computational approaches. Furthermore, given that the treatment experiments of HCC1143 with Trametinib and BEZ235 were likely repeated, the authors have not provided evidence that the cell lines subjected to WES did show similar phenotypic shifts previously observed.”

The new figures (Supplementary Fig. 6b-d) include phase and immunofluorescent images showing the expected phenotypic shifts in the treated cells used for WES, as well as panels showing the sequencing statistics and sequencing coverage distribution in the different experimental conditions. We have also added a new supplementary table (Supplementary Table 1) showing all the variants identified between experimental conditions with MuTect analysis. This information includes the variant allele frequency, genomic location, associated gene, and if applicable, reasons to suspect that the variant call is due to technical error.

All variants showed an intronic, intergenic, or non-coding genomic location with no predicted functional impact on their respective gene product. In particular, as per the reviewer’s query, none of the variants involved known cancer genes. Furthermore, all variants showed low variant allele frequencies (avg. =14%), which would not be expected in a clonal selection model and is far below the changes in phenotypic composition observed by immunofluorescence (e.g.

K14^{hi} cells go from 23% in the DMSO condition to 81% following Trametinib treatment). Lastly, most variants occurred in particularly error-prone, low-coverage regions suggesting that they were sequencing artifacts. Together, these findings led us to conclude and to state that genomic selection plays a “minimal” role in the observed differentiation-state enrichments by Trametinib and BEZ235. We realize our experiments do not rule out the possibility that genome selection plays a small role in the observed state changes, so we have softened our language in the manuscript to conclude that state-transition is the “primary” driver of drug-induced differentiation-state enrichment in our experiments.

All changes in the manuscript have been highlighted in blue for easy tracking.

We have also improved access to the raw sequencing files which can be accessed on the Sequence Read Archive by the following link: <https://www.ncbi.nlm.nih.gov/sra/SRP125560>, and have updated our data availability statement accordingly.

Thank you for the time and effort in reviewing our manuscript. We appreciate your comments, and believe they have helped improve our work.

Reviewers' comments:

Reviewer #2 (Remarks to the Author):

This reviewer is puzzled as to why the authors are so reluctant to provide the results of the sequencing analysis in a meaningful way. Now that the details have been provided, it is blatantly clear that they minimized the chances of finding genomics differences according to treatment rather than performing an unbiased experiment to confirm or refute their hypothesis.

It behooves the authors to provide the minimal analysis of the whole exome sequencing data generated, calling not only the SNVs (which was done in this draft of the manuscript), but also the small indels and the copy number alterations. The latter is particularly relevant, as differences in copy numbers of specific genes have been shown to mediate resistance to specific small molecule inhibitors through clonal selection (Xue et al. Nat Med. 2017 Aug;23(8):929-937).

Based on its current form, I still cannot recommend the acceptance of this manuscript, as the authors have failed to rule out an obvious alternative explanation for their findings.

I would be willing to re-review a revised version of the manuscript should the authors provide the analyses and results requested.

Dear Reviewer,

We thank you for your comments on our manuscript “Differentiation-state plasticity is a targetable resistance mechanism in basal-like breast cancer”. We have addressed your concerns with the addition of new experiments, analyses, and supplementary data in line with your suggestions as detailed in our point-by-point response below.

Reviewer #1:

The sequencing data presented in this manuscript, in its current form, does not address the comments this reviewer previously made. It is absolutely essential that the authors provide a detailed account of the sequencing findings.

In response to this concern, we have now performed additional analyses to examine indels within our WES data as well as new whole genome sequencing to analyze copy number alterations. We have included all requested supplementary data files, and have uploaded the raw sequencing files to the Sequence Read Archive. Data are presented in Supplementary Figure 7, Supplementary Table 1, Supplementary Table 2, and Supplementary Data Set 4, and are described and highlighted in the results section. Details of new methods are included in Methods and Supplementary Methods. All raw sequencing files can be accessed under accession numbers SRP125560 (WES) and SRP144106 (LP-WGS).

This reviewer is also unsure if small indels were adequately detected with the approaches used.

In response to this concern we have now applied GATK4 MuTect2 analysis software to our bam files from the WES experiment. The analysis identified potential indels between HCC1143 treatments, and these are now presented along with the SNVs in Supplementary Table 1. Similar to the identified variants, most identified indels (17/19) were located in intergenic, intronic, or noncoding RNA regions with no predicted functional gene impact. All identified indels, similar to the SNV calls, also had low variant allele frequencies (VAFs), below the frequency of changes in phenotypic state observed by imaging. The two deletions found in nonsynonymous gene coding or regulatory regions were a frameshift deletion in CD93 (12% VAF) and a 3'UTR deletion in TRMT1L (11% VAF). In both cases, the deletions were shared by two out of the three treatments (DMSO and Trametinib for CD93, and Trametinib and BEZ235 for TRMT1L). We believe that both of these are likely alignment artifacts caused by repetitive sequence regions (nine x CAG/Leucine repeat in CD93 and 14 x A repeat in TRMT1L).

Regardless of the validity of these calls, we investigated the expression of CD93 and TRMTL1 in our RNA-sequencing data and found that CD93 is not expressed in this cell line under control or treatment conditions and that TRMT1L expression does not significantly change with treatment, indicating these potential indels have minimal or no impact on cell response (TRMTL1 gene expression is now shown in Supplementary Fig. 7d). VCF files for all variant and indel calls in pairwise comparisons of all treatment conditions are included in Supplementary Dataset 4.

In addition, no analysis of the copy number alterations was provided, which can also constitute a source of heterogeneity and a mechanism of selection.... In addition, a detailed analysis of the differences in copy number ought to be provided.

We have now performed a new low pass whole genome sequencing (LP-WGS) experiment to address this concern. Replicate DNA isolates from the WES experiment were pooled by condition and analyzed by 0.5X average coverage WGS. We performed copy number alteration analysis on the resultant data using ichorCNA software (Adalsteinsson et al., Nat. Comm., 2017) and have presented the CNA plots for each treatment in Supplementary Fig. 7e. These plots show concordance between treatments and control and do not support the selection of cells with distinct copy number profiles during treatment.

A complete list of somatic variants (SNVs and indels) and their annotations is required, not only a difference between the different treatment conditions with minimal annotation of the alterations identified. The mutations in common should also be provided in the form of a supplementary dataset.

The complete list of the SNVs and indels that were identified in pairwise comparisons between all treatment conditions is available as Supplemental Table 1 with annotations of location, genomic changes and protein changes. We also have now compared each treatment condition to the normal control HCC1143-BL. All common and unique SNVs and indels from this analysis, along with annotation, are now presented in Supplementary Table 2. In this analysis, most of the variants identified between the tumor cell line and normal were shared between all treatments. Six variants were found in only one or two treatments, but similar to the pairwise comparison described above, all of these variants were in intronic or intergenic regions and had low VAFs. Again, all VCF files for this analysis are available in Supplementary Dataset 4.

The analysis methods for the SNVs are described in sufficient detail, however there is no description of the details of the indel detection or copy number alteration analysis.

The description of indel detection, LP-WGS, and CNA detection have all been added to the Methods and Supplementary Methods sections of our manuscript, with citations of all analyses used and links for software access.

The authors should include a supplementary data file with the VCF or MAF for each of the sequencing experiments.

These files are now included in Supplementary Dataset 4. This dataset includes a table describing all provided files, VCF files for SNVs and indels identified in the pairwise comparisons of treatments, VCF files for SNVs and indels identified in comparison of treatment samples to the normal HCC1143-BL sample, and the log2 ratio segmentation files from the LP-WGS analysis.

The priority of this manuscript would be greatly enhanced if the authors were in a position of providing the same rigor and detail for the genomics analyses, as they did for the other aspects of this manuscript.

We believe that with the addition of these new genomic analyses we have sufficiently addressed this concern.

All changes in the manuscript have been highlighted in yellow for easy tracking.

Thank you for the time and effort in reviewing our manuscript. We appreciate your comments, and believe they have helped improve our work.

With kind regards,

Rosalie Sears, Ph.D.
Professor, Department of Molecular and Medical Genetics
Co-Director, Brenden-Colson Center for Pancreatic Care
Knight Cancer Institute

Joe W. Gray, Ph.D.
Gordon Moore Endowed Chair and Professor, Department of Biomedical Engineering
Director, OHSU Center for Spatial Systems Biomedicine
Associate Director for Biophysical Oncology, Knight Cancer Institute